# Dynamical Properties of Dense Associative Memory

**Kazushi Mimura**
Hiroshima City University, Hiroshima, Japan
RIKEN AIP, Tokyo, Japan
mimura@hiroshima-cu.ac.jp
kazushi.mimura@a.riken.jp

**Jun'ichi Takeuchi**
Graduate School of Info. Sci. and Elect. Eng.
Kyushu University, Fukuoka, Japan
tak@inf.kyushu-u.ac.jp

**Yuto Sumikawa**
Department of Physics
The University of Tokyo, Tokyo, Japan
sumikawa-yuto@g.ecc.u-tokyo.ac.jp

**Yoshiyuki Kabashima**
Institute for Physics of Intelligence
The University of Tokyo, Tokyo, Japan
kaba@phys.s.u-tokyo.ac.jp

**Anthony C. C. Coolen**
Saddle Point Science Europe and DCN Donders Institute,
Radboud University, Nijmegen, Netherlands
a.coolen@science.ru.nl

## Abstract

Dense associative memory, a fundamental instance of modern Hopfield networks, can store a large number of memory patterns as equilibrium states of recurrent networks. While the stationary-state storage capacity has been investigated, its dynamical properties have not yet been discussed. In this paper, we analyze the dynamics using an exact approach based on generating functional analysis. We show results on convergence properties of memory retrieval, such as the convergence time and the size of the attraction basins. Our analysis enables a quantitative evaluation of the convergence time and the storage capacity of dense associative memory, which is useful for model design. Unlike the traditional Hopfield model, the retrieval of a pattern does not act as additional noise to itself, suggesting that the structure of modern networks makes recall more robust. Furthermore, the methodology addressed here can be applied to other energy-based models, and thus has the potential to contribute to the design of future architectures.

## 1 Introduction

### 1.1 Background

Dense associative memory (Krotov & Hopfield, 2016), a model for storing binary patterns, was proposed and shown to significantly improve the storage capacity of the traditional Hopfield model (Hopfield, 1982). While it can be regarded as a rediscovery of the many-body Hopfield model (Gardner, 1987; Abbott & Arian, 1987), it exhibits slightly different properties. On the other hand, extensions of dense associative memory, such as the Hopfield layer, have been actively developed to enable dense associative memory to store real-valued patterns (Demircigil et al., 2017; Ramsauer et al., 2021). Hopfield models with such large memory capacities, including these variants, are referred to as modern Hopfield networks, which have gained increasing attention and have even inspired Transformer architectures (Hoover et al., 2023).

The equilibrium properties of the Hopfield layer have been analytically studied, including evaluations of its storage capacity (Lucibello & Mézard, 2024). Since the Hopfield layer can reach a near-equilibrium state in almost a single update step, its dynamical properties have not been considered a significant issue. In contrast, dense associative memory, like the traditional Hopfield model, requires some iterative updates to reach a stationary state. However, its dynamical behavior has not

been investigated so far. As a result, fundamental aspects such as the attraction basin, namely, how far from a stored pattern can the initial state be for the system to still successfully recall it, still remain unclear.

While the dynamical properties of dense associative memory have not been investigated, those of the traditional Hopfield model have been extensively analyzed. In this paper, we analyze the dynamical properties of dense associative memory using generating functional analysis, an asymptotic theory in the large-system limit, which has been widely used in those studies.

## 1.2 CONTRIBUTIONS

Our main contributions are as follows:

- Asymptotically exact dynamical analysis. – We provide, for the first time, an asymptotically exact analysis of the dynamics of dense associative memory in the large-system limit using generating functional analysis (GFA).

- Quantitative characterization of convergence. – Our analysis yields explicit results on convergence properties of memory retrieval, including convergence time and the size of attraction basins, thereby enabling quantitative evaluation of stability and storage capacity.

- Novel insight into robustness of modern Hopfield networks. – We demonstrate that, unlike the traditional Hopfield model, retrieval does not introduce additional self-noise, suggesting that the architecture of modern networks makes recall more robust.

- General methodology for energy-based models. – The proposed framework is not limited to dense associative memory. It can be applied to other energy-based models, providing theoretical tools for the design of robust and scalable architectures.

## 1.3 RELATED WORKS

Gardner and Abbott independently introduced a Hopfield model with many-body interactions, which is essentially equivalent to dense associative memory, and analyzed its equilibrium properties using the replica method to evaluate its storage capacity. The difference between their models and the dense associative memory proposed by Krotov and Hopfield lies in the presence or absence of self-coupling terms. While this difference does not affect the order of the storage capacity, it does influence the constant factor. Additionally, Lucibello and Mézard analyzed the equilibrium properties of the Hopfield layer using the replica method and obtained its storage capacity (Lucibello & Mézard, 2024). So far, no analysis of the dynamical behavior of the modern Hopfield model has been reported. On the other hand, there has been extensive research on the dynamics of the traditional Hopfield models. For example, the papers (Gardner et al., 1987; Crisanti & Sompolinsky, 1987; 1988; Rieger et al., 1988) and our previous papers provide exact analysis based on GFA.

## 1.4 RELATED WORKS

Gardner and Abbott independently introduced a Hopfield model with many-body interactions, which is essentially equivalent to dense associative memory, and analyzed its equilibrium properties using the replica method to evaluate its storage capacity. The difference between their models and the dense associative memory proposed by Krotov and Hopfield lies in the presence or absence of self-coupling terms. While this difference does not affect the order of the storage capacity, it does influence the constant factor. Additionally, Lucibello and Mézard analyzed the equilibrium properties of the Hopfield layer using the replica method and obtained its storage capacity (Lucibello & Mézard, 2024). So far, no analysis of the dynamical behavior of the modern Hopfield model has been reported. On the other hand, there has been extensive research on the dynamics of the traditional Hopfield models. For example, the papers (Gardner et al., 1987; Crisanti & Sompolinsky, 1987; 1988; Rieger et al., 1988; Düring et al., 1998b; Coolen, 2000; Mimura et al., 2004; Mimura & Coolen, 2009) provide exact analysis based on GFA.

## 2 PRELIMINALIES

### 2.1 NOTATIONS

Throughout this paper, vectors are denoted by boldface, e.g., $\boldsymbol{x}$, and are assumed to be column vectors unless otherwise stated. $x^{(t)}$ and $(\boldsymbol{x})^{(t)}$ represent the $t$-th element of the vector $\boldsymbol{x}$. Matrices are denoted by an upper case symbol, e.g., $A$, and $A^{\top}$ denotes the transpose of a matrix $A$.

### 2.2 DENSE ASSOCIATIVE MEMORY

The dense associative memory is one of the recurrent neural network models to store and recall a large number of patterns as fixed points of the dynamics. The energy of dense associative memory is given by

$$H = -\sum_{\mu=1}^{M} F\left(\sum_{i=1}^{N} \xi_i^{\mu} h_i\right), \tag{1}$$

where $h_i \in \{\pm 1\}$ denotes the state of the $i$-th unit, and $\xi_i^{\mu}$ denotes the $i$-th element of the $\mu$-th pattern. Each $\xi_i^{\mu}$ independently takes the value $\pm 1$ with equal probability $1/2$. Introducing a nonlinear function $F$, such as a power function, makes memory patterns become deeper minima in the energy landscape and reduces interference between different memory patterns. This is because the nonlinearity suppresses weaker overlaps during the recall process. In this paper, we restrict ourselves to the case

$$F(x) = \frac{x^n}{2N^{n-1}}. \tag{2}$$

It should be noted that since the coefficient $1/(2N^{n-1})$ does not affect the performance, so it is equivalent to setting $F(x) = x^n$. The update rule is defined by the difference of two energies before and after state transitions. We keep only the leading term in the argument of the sgn function in the update rule, which gives

$$h_i^{(t+1)} = \text{sgn}\left[\sum_{\mu=1}^{M} F\left(+\xi_i^{\mu} + \sum_{j\neq i}^{N} \xi_j^{\mu} h_j\right) - \sum_{\mu=1}^{M} F\left(-\xi_i^{\mu} + \sum_{j\neq i}^{N} \xi_j^{\mu} h_j\right)\right] \tag{3}$$

$$= \text{sgn}\left[\sum_{\mu=1}^{M} \xi_i^{\mu} n\left(\frac{1}{N} \sum_{j\neq i}^{N} \xi_j^{\mu} h_j\right)^{n-1} + \text{(small order terms)}\right], \tag{4}$$

where $\text{sgn}(x)$ denotes the sign function that takes $1$ if $x \geq 0$, and $-1$ otherwise. In the case of $n = 2$ the network reduces to the parallel dynamics version of traditional Hopfield model, i.e., $h_i^{(t+1)} = \text{sgn}\left(\sum_{j=1}^{N} J_{ij} h_j^{(t)}\right)$ and $J_{ij} = \frac{1}{N} \sum_{\mu=1}^{M} \xi_i^{\mu} \xi_j^{\mu}$. By focusing on the leading terms for a given function $F$, we can treat arbitrary activation functions. However, note that in the case of the exponential function, all terms in the power-series expansion have the same order.

### 2.3 OUTLINE OF GENERATING FUNCTIONAL ANALYSIS

We apply the generating functional analysis (GFA) to investigate dynamical properties of the dense associative memory. GFA has been applied to the model which is described using realizations of random variables (DeDominicis, 1978). This method allows us to analyze the asymptotic dynamical behavior in the infinitely large system, using the generating functional, which is the dynamical equivalent of the characteristic function in statistics.

In GFA formalism, we consider the joint probability distribution over the states of all units at all time steps, from the start of the iteration up to some prescribed time, which can be taken sufficiently large. This joint probability is referred to as the *path probability*. From the path probability, we can calculate various expectation values such as the *overlap*, which is the direction cosine between the states of the units and the memory pattern being recalled via the *generating functional* which can be regarded as an analogue of the characteristic function in statistics. Table 1 shows the representative analysis for dynamical properties of traditional and modern Hopfield models.

| Paper | Model | Method | Update | Retarded SI |
|---|---|---|---|---|
| Amari & Maginu (1988) | traditional | S/N | parallel | ignored |
| Okada (1995) | traditional | hierarchical S/N | parallel | ignored |
| Rieger et al. (1988) | traditional | generating functional | asynchronous | treated |
| Coolen & Sherrington (1994) | traditional | dynamical replica | asynchronous | treated |
| Düring et al. (1998a) | traditional | generating functional | parallel | treated |
| this paper | modern | generating functional | parallel | treated |

Table 1: Relationship to existing dynamical analyses for Hopfield models.

## 3 ANALYSIS

First, the path probability is defined and used to describe the generating functional, after which the expectation over the memory patterns appearing in the generating functional is evaluated.

### 3.1 PATH PROBABILITY

Let vectors $\boldsymbol{h}^{(t)} = (h_1^{(t)}, \cdots, h_N^{(t)})^\top \in \{\pm 1\}^N$ be the states of all units at time $t$ and let the initial state be $\boldsymbol{h}^{(0)}$. The updating rule, obtained by retaining only the leading term, is expressed as follows:

$$h_i^{(t+1)} = \text{sgn}\,(u_i^{(t)}), \tag{5}$$

$$u_i^{(t)} = \sum_{\mu=1}^M \xi_i^\mu n \left( \frac{1}{N} \sum_{j \neq i}^N \xi_j^\mu h_j^{(t)} \right)^{n-1} + \theta_i^{(t)}, \tag{6}$$

for all $i \in \{1, \cdots, N\}$ and $t \in \{0, \cdots, T-1\}$. The variable $u_i^{(t)}$ is referred to as a *local field*. The parameter $\theta_i^{(t)}$ is called an *external field* (or an *threshold*). The dynamics of the system are characterized by how the output of each unit changes in response to infinitesimal variations in the local field. To evaluate such changes, the external field $\{\theta_i^{(t)}\}$ are introduced. The average derivative of the outputs with respect to the external field is referred to as the response function, which serves as one of fundamental measures for describing the dynamics. After evaluating the response function, all $\{\theta_i^{(t)}\}$ are set to zero.

In this paper, we consider parallel dynamics, in which the states of all units are updated simultaneously. The updating rule of the dense associative memory for the variable $\boldsymbol{h}^{(t+1)}$ at time $t$ can be given by the following probability distribution:

$$p[\boldsymbol{h}^{(t+1)}|\boldsymbol{h}^{(t)}] = \prod_{i=1}^N \delta[h_i^{(t+1)}; \text{sgn}\,(u_i^{(t)})], \tag{7}$$

where $\delta[m; n]$ denotes the Kronecker's delta that takes 1 if $m = n$ and 0 otherwise. This dynamics represents Markovian dynamics. The path probability $p[\boldsymbol{h}^{(0)}, \cdots, \boldsymbol{h}^{(T)}]$ is given as the products of the probability distribution of the updating rule:

$$p[\boldsymbol{h}^{(0)}, \cdots, \boldsymbol{h}^{(T)}] = p[\boldsymbol{h}^{(0)}] \prod_{t=0}^{T-1} p[\boldsymbol{h}^{(t+1)}|\boldsymbol{h}^{(t)}], \tag{8}$$

where $p[\boldsymbol{h}^{(0)}] = \prod_{i=1}^N p[h_i^{(0)}]$ denotes the initial state distribution. Since the same memory patterns are included at every time step, the states of the units at different times are correlated.

### 3.2 GENERATING FUNCTIONAL

The path probability depends on all memory patterns $\boldsymbol{\xi}^1, \cdots, \boldsymbol{\xi}^M$. We define the generating functional as follows.

**Definition 1.** *The generating functional $\bar{Z}[\boldsymbol{\psi}]$ is defined as*

$$\bar{Z}[\boldsymbol{\psi}] = \mathbb{E}_{\boldsymbol{\xi}^1,\cdots,\boldsymbol{\xi}^M}\left[\sum_{\boldsymbol{h}^{(0)},\cdots,\boldsymbol{h}^{(T)}\in\{\pm1\}^N} p[\boldsymbol{h}^{(0)},\cdots,\boldsymbol{h}^{(T)}]\exp\left(-i\sum_{t=0}^T \boldsymbol{h}^{(t)}\cdot\boldsymbol{\psi}^{(t)}\right)\right], \quad (9)$$

*where we have introduced the generating variables $\boldsymbol{\psi}^{(t)} = (\psi_1^{(t)},\cdots,\phi_N^{(t)})^\top \in \mathbb{R}^N$ and we write $\boldsymbol{\psi} = (\boldsymbol{\psi}^{(0)},\cdots,\boldsymbol{\psi}^{(T)})$ for shorthand.*

Here, $i$ denotes the imaginary unit. i.e., $i = \sqrt{-1}$. We here assumed that the generating functional is self-averaging, namely, in the large-system limit, i.e., $N$ is sufficiently large, the generating functional is concentrated on its average over the memory patterns $\boldsymbol{\xi}^1,\cdots,\boldsymbol{\xi}^M$ and the typical behaviour of the model only depends on the statistical properties of the memory patterns. In GFA, the expectation values of interest are calculated from derivatives with respect to some elements of the generating variables, e.g.,

$$\lim_{\boldsymbol{\psi}\to\boldsymbol{0}}\frac{\partial\bar{Z}[\boldsymbol{\psi}]}{\partial\psi_i^{(t)}} = \mathbb{E}_{\boldsymbol{\xi}^1,\cdots,\boldsymbol{\xi}^M}[\langle -ih_i^{(t)}\rangle], \quad (10)$$

$$\lim_{\boldsymbol{\psi}\to\boldsymbol{0}}\frac{\partial^2\bar{Z}[\boldsymbol{\psi}]}{\partial\psi_i^{(t)}\partial\psi_i^{(t')}} = \mathbb{E}_{\boldsymbol{\xi}^1,\cdots,\boldsymbol{\xi}^M}[\langle -h_i^{(t)}h_{i'}^{(t')}\rangle], \quad (11)$$

$$\lim_{\boldsymbol{\psi}\to\boldsymbol{0}}\frac{\partial^2\bar{Z}[\boldsymbol{\psi}]}{\partial\psi_i^{(t)}\partial\theta_i^{(t')}} = \mathbb{E}_{\boldsymbol{\xi}^1,\cdots,\boldsymbol{\xi}^M}[\langle -i\frac{\partial h_i^{(t)}}{\partial\theta_{i'}^{(t')}}\rangle], \quad (12)$$

where $\boldsymbol{\psi}\to\boldsymbol{0}$ denotes $\psi_i^{(t)}\to 0$ for all $i$ and $t$, and the bracket $\langle\cdots\rangle$ denotes the average over the path probability, i.e., $\langle(\cdots)\rangle = \sum_{\boldsymbol{h}^{(0)},\cdots,\boldsymbol{h}^{(T)}\in\{\pm1\}^N} p[\boldsymbol{h}^{(0)},\cdots,\boldsymbol{h}^{(T)}](\cdots)$. Introducing the definition of the local field using the Dirac delta function, the generating functional can be rewritten as follows:

$$\bar{Z}[\boldsymbol{\psi}] = \mathbb{E}_{\boldsymbol{\xi}^1,\cdots,\boldsymbol{\xi}^M}\left[\sum_{\boldsymbol{h}^{(0)},\cdots,\boldsymbol{h}^{(T)}}\int_{\mathbb{R}^T}d\boldsymbol{u}\, p[\boldsymbol{h}^{(0)}]\left(\prod_{t=0}^{T-1}\prod_{i=1}^N\delta[h_i^{(t+1)};\text{sgn}(u_i^{(t)})]\right)e^{-i\sum_{t=0}^T\boldsymbol{h}^{(t)}\cdot\boldsymbol{\psi}^{(t)}}\right.$$

$$\left.\times\left(\prod_{t=0}^{T-1}\prod_{i=1}^N\delta\left(u_i^{(t)} - \sum_{\mu=1}^M\xi_i^\mu\left(\frac{1}{N}\sum_{j\neq i}^N\xi_j^\mu h_j^{(t)}\right)^{n-1} - \theta_i^{(t)}\right)\right)\right]. \quad (13)$$

Assuming that the pattern $\boldsymbol{\xi}^1$ is being recalled, we separate the local field in the generating functional into a signal term including the recalling pattern and a noise term including other patterns $\boldsymbol{\xi}^2,\cdots,\boldsymbol{\xi}^M$. Using the Fourier integral form of Dirac delta function, the generating functional becomes

$$\bar{Z}[\boldsymbol{\psi}] = \sum_{\boldsymbol{h}^{(0)},\cdots,\boldsymbol{h}^{(T)}}\int_{\mathbb{R}^T}d\boldsymbol{u}\delta\hat{\boldsymbol{u}}\, p[\boldsymbol{h}^{(0)}]\left(\prod_{t=0}^{T-1}\prod_{i=1}^N\delta[h_i^{(t+1)};\text{sgn}(u_i^{(t)})]\right)e^{-i\sum_{t=0}^T\boldsymbol{h}^{(t)}\cdot\boldsymbol{\psi}^{(t)}}$$

$$\times\exp\left[i\sum_{t=0}^T\sum_{i=1}^N\hat{u}_i^{(t)}(u_i^{(t)} - \theta_i^{(t)})\right]\left(\mathbb{E}_{\boldsymbol{\xi}^1}\exp\left[-i\sum_{t=0}^{T-1}\sum_{i=1}^N\hat{u}_i^{(t)}\xi_i^1\, n\left(\frac{1}{N}\sum_{j\neq i}^N\xi_j^1 h_j^{(t)}\right)^{n-1}\right]\right)$$

$$\times\left(\mathbb{E}_{\boldsymbol{\xi}^2,\cdots,\boldsymbol{\xi}^M}\exp\left[-i\sum_{t=0}^{T-1}\sum_{i=1}^N\sum_{\mu=2}^M\hat{u}_i^{(t)}\xi_i^\mu\, n\left(\frac{1}{N}\sum_{j\neq i}^N\xi_j^\mu h_j^{(t)}\right)^{n-1}\right]\right) \quad (14)$$

Only the last part involves all non-recalled patterns $\boldsymbol{\xi}^2,\cdots,\boldsymbol{\xi}^M$. By straightforward calculation of the expectation over these patterns, the generating functional is found to depend on five types of averages. Accordingly, we introduce the following macroscopic parameters:

$$m^{(t)} = \frac{1}{N}\sum_{i=1}^N\xi_i h_i^{(t)}, \quad k^{(t)} = \frac{1}{N}\sum_{i=1}^N\xi_i\hat{u}_i^{(t)},$$

$$q^{(t,t')} = \frac{1}{N} \sum_{i=1}^{N} h_i^{(t)} h_i^{(t')}, \quad Q^{(t,t')} = \frac{1}{N} \sum_{i=1}^{N} h_i^{(t)} \hat{u}_i^{(t')}, \quad K^{(t,t')} = \frac{1}{N} \sum_{i=1}^{N} h_i^{(t)} \hat{u}_i^{(t')}, \qquad (15)$$

into the generating functional using the Dirac delta function, where the functions $m^{(t)}$ is referred to as the *overlap*. The generating functional can be calculated as follows.

**Lemma 1.** *By averaging over the memory patterns, the generating functional is given by*

$$\bar{Z}[\boldsymbol{\psi}] = \int d\boldsymbol{m} d\hat{\boldsymbol{m}} d\boldsymbol{k} d\hat{\boldsymbol{k}} d\boldsymbol{q} d\hat{\boldsymbol{q}} d\boldsymbol{Q} d\hat{\boldsymbol{Q}} d\boldsymbol{K} d\hat{\boldsymbol{K}} \ \exp\left[N(\Psi + \Phi + \Omega) + O(\log N)\right], \qquad (16)$$

*where*

$$\Psi = i \sum_{t=0}^{T-1} \left\{ \hat{m}^{(t)} m^{(t)} + \hat{k}^{(t)} k^{(t)} - k^{(t)} n (m^{(t)})^{n-1} \right\}$$

$$+ i \sum_{t=0}^{T-1} \sum_{t'=0}^{T-1} \left\{ \hat{q}^{(t,t')} q^{(t,t')} + \hat{Q}^{(t,t')} Q^{(t,t')} + \hat{K}^{(t,t')} K^{(t,t')} \right\}, \qquad (17)$$

$$\Phi = \frac{1}{N} \log \sum_{\boldsymbol{h}} \int d\boldsymbol{u} d\hat{\boldsymbol{u}} \ p[\boldsymbol{h}^{(0)}] \left( \prod_{t=0}^{T-1} \prod_{i=1}^{N} \delta[h_i^{(t+1)}; \mathrm{sgn}(u_i^{(t)})] \right)$$

$$\times E_{\boldsymbol{\xi}} \exp\left[ -i \sum_{t=0}^{T-1} \sum_{t'=0}^{T-1} \sum_{i=1}^{N} \left\{ \hat{q}^{(t,t')} h_i^{(t)} h_i^{(t')} + \hat{Q}^{(t,t')} \hat{u}_i^{(t)} \hat{u}_i^{(t')} + \hat{K}^{(t,t')} h_i^{(t)} \hat{u}_i^{(t')} \right\} \right.$$

$$\left. + i \sum_{t=0}^{T-1} \sum_{i=1}^{N} \hat{u}_i^{(t)} \{ u_i^{(t)} - \hat{k}^{(t)} - \theta_i^{(t)} \} - i \sum_{t=0}^{T-1} \sum_{i=1}^{N} h_i^{(t)} \hat{m}^{(t)} \xi_i - i \sum_{t=0}^{T-1} \sum_{i=1}^{N} h_i^{(t)} \psi_i^{(t)} \right], \qquad (18)$$

$$\Omega = -\frac{1}{2} n^2 \frac{M}{N^{n-1}} \sum_{t=0}^{T-1} \sum_{t'=0}^{T-1} \left\{ (n-1)^2 \left[ \sum_{k=0}^{n-2} A(n-2,k) (q^{(t,t')})^k \right] K^{(t',t)} K^{(t,t')} \right.$$

$$\left. + \left[ \sum_{k=0}^{n-1} A(n-1,k) (q^{(t,t')})^k \right] Q^{(t',t)} \right\} + O(N^{-1}). \qquad (19)$$

*where* $A(\ell, k) = \binom{\ell}{k}^2 k! \ B(\ell-k)^2$, *and* $B(m) = \mathbf{1}_{m:even} \ (m-1)!!$.

A proof is given in Appendix A. Here, $\mathbf{1}_{\mathrm{condition}}$ denotes the indicator function that takes $1$ if the condition is true, and $0$ otherwise. It can be obtained by evaluating the leading terms after taking the expectation over the memory patterns, using combinatorial arguments. The order of the number of memory patterns is determined by the balance between the magnitude of the signal originating from the retrieved pattern and that of the noise originating from the non-retrieved patterns. From the analysis in Lemma 1, the number of memory patterns $M$ is required to scale as $M = O(N^{n-1})$ for non-trivial analysis. This corresponds to the generating functional being of order $e^{O(N)}$. Therefore, we set

$$M = \alpha_n N^{n-1}. \qquad (20)$$

Further details are given in Appendix A.

The generating functional is dominated by a saddle-point in the large-system limit. Averaging over the random variables, we will move to a saddle-point problem (Copson, 1965) in the limit $N \to \infty$. The saddle point condition gives values of the macroscopic parameters. Hereafter, we choose the factorised distribution $p[\boldsymbol{h}^{(0)}] = \prod_{i=1}^{N} p[h_i^{(0)}] = \prod_{i=1}^{N} \{ \frac{1}{2}(1 + m^{(0)}) \delta[h^{(0)}; \xi_i] + \frac{1}{2}(1 - m^{(0)}) \delta[h^{(0)}; -\xi_i] \}$ as an initial state distribution, where $m^{(0)}$ denotes an *initial overlap*. The factorised initial overlap allows the generating functional to decompose into independent single-unit contributions.

## 4 MAIN RESULTS

The behavior of this model differs significantly between the case $n = 2$ and the case $n \geq 3$. Since the case $n = 2$ has already been extensively studied, we focus only on the case $n \geq 3$ in this paper. GFA provides an exact solution as an asymptotic analysis in the large-system limit $N \to \infty$. Using the saddle point method to evaluate the integral in the averaged generating functional, one can obtain the following proposition.

**Proposition 1.** *For a given initial state distribution $p[h^{(0)}]$ and $n \geq 3$, the overlap $m^{(t)}$, the correlation function $C^{(t,t')}$, and the response function $G^{(t,t')}$ are given by*

$$m^{(t)} = \langle\!\langle \xi h^{(t)} \rangle\!\rangle, \quad C^{(t,t')} = \langle\!\langle h^{(t)} h^{(t')} \rangle\!\rangle, \quad G^{(t,t')} = \mathbf{1}_{t>t'} \frac{\partial \langle\!\langle h^{(t)} \rangle\!\rangle}{\partial \theta^{(t')}}, \tag{21}$$

*where $\langle\!\langle f(\boldsymbol{h}) \rangle\!\rangle$ denotes the average defined as*

$$\langle\!\langle f(\boldsymbol{h}) \rangle\!\rangle = \mathbb{E}_\xi \int \mathcal{D}\boldsymbol{v} \sum_{\boldsymbol{h}} f(\boldsymbol{h}) p[h^{(0)}] \prod_{t=0}^{T-1} \delta\left[ h^{(t+1)}; \mathrm{sgn}\left( \xi n (m^{(t)})^{n-1} + (\Gamma \boldsymbol{h})^{(t)} + v^{(t)} + \theta^{(t)} \right) \right], \tag{22}$$

*which is referred to as the effective path measure. The random vector $\boldsymbol{v}$ follows a multivariate normal distribution with mean $\mathbf{0}$ and covariance matrix $R = (R^{(t,t')})$, where the $(t,t')$-element is*

$$R^{(t,t')} = n^2 \alpha_n \sum_{k=0}^{n-1} A(n-1, k) (C^{(t,t')})^k. \tag{23}$$

*The matrix $\Gamma$ is given by $\Gamma = D \circ G$. The $(t,t')$-elements of $D$ and $G$ are $D^{(t,t')}$ and $G^{(t,t')}$, respectively. Each element of the matrix $D = (D^{(t,t')})$ is defined as*

$$D^{(t,t')} = n^2 (n-1)^2 \alpha_n \sum_{k=0}^{n-2} A(n-2, k) (C^{(t,t')})^k. \tag{24}$$

*The operator $\circ$ denotes the Hadamard (elementwise) product.*

The proof sketch is given in Appendix B. The term $(\Gamma \boldsymbol{h})^{(t)}$ in the effective path measure represents a *retarded self-interaction*. The retarded self-interaction means the magnitude of the influence that returns to a unit itself after propagating through other units. Due to this retarded self-interaction, the state at the next time step depends in a complex way on the past states. The covariance matrix $R$ has no explicit dependence on the overlap $m$ or the response function $G$; it depends on them only implicitly through the correlation function $C^{(t,t')}$, which is determined self-consistently with $m$ and $G$. In particular, the diagonal element $R^{(t,t)}$ is independent of $m$ and $G$. This suggests that the phenomenon, in which the system begins to recall correctly but eventually fails to complete it, becomes less likely to occur.

## 5 DISCUSSION

### 5.1 NUMERICAL ANALYSIS AND COMPUTER SIMULATIONS

In this paper we considered Krotov's dense associative memory. Since the noise variance depends on $n$, we normalize the constant $\alpha_n$ by setting $\alpha'_n = (2n-3)!! \alpha_n$, where $\alpha'_n$ is referred to as the *loading rate*. The *storage capacity* $\alpha'_{c,n}$ is defined as the largest loading rate at which the overlap remains positive.

The result of Proposition 1 can be numerically analyzed using the Monte Carlo method. Figure 1 shows the numerical analysis for the case $n = 3$, together with the results of computer simulations. In Fig. 1 (a) – (d), the graphs on the left display the simulation results for $N = 1024$ with 100 trials. The vertical axis represents the overlap, while the horizontal axis represents the number of iteration steps. The graphs on the right in Fig. 1 (a) – (d) correspond to the numerical analysis of the overlap based on Proposition 1. Although finite-size effects become significant near the basin

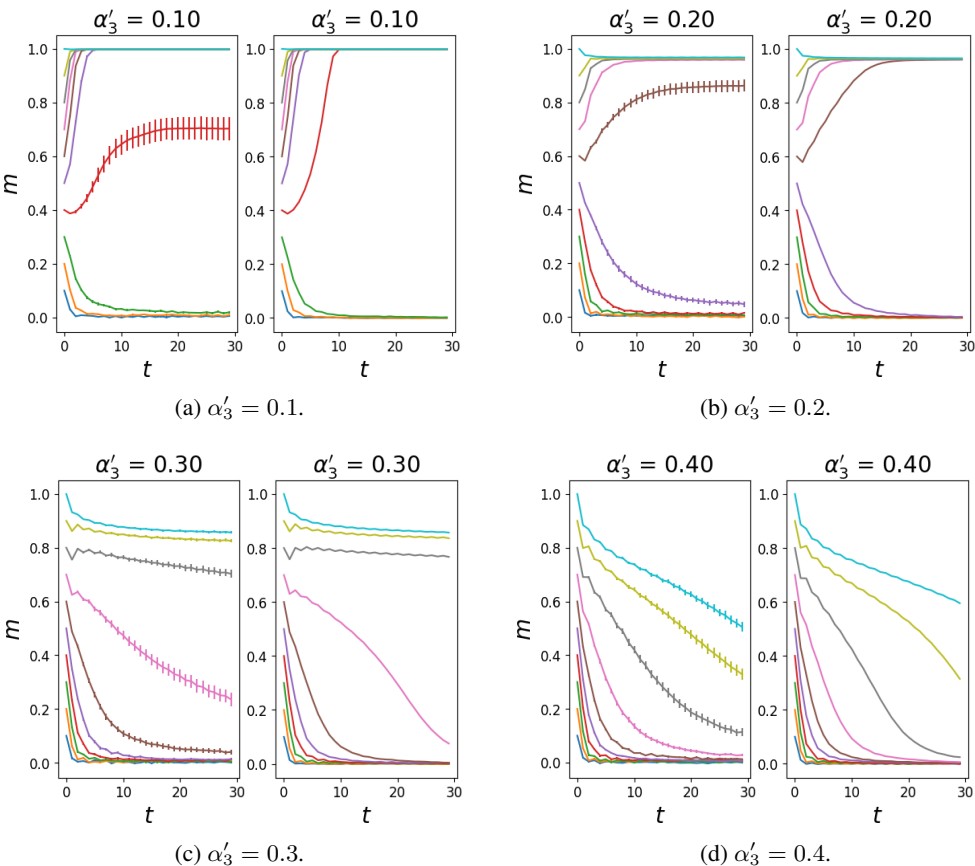

Figure 1: Recalling process of Krotov's dense associative memory with $F(x) = x^n$ and $n = 3$. Left: computer simulations, 100 trials, $N = 1024$. Right: theory.

of attraction, it can be confirmed that the theoretical values agree well with the simulation results even for relatively small-scale experiments. It can be theoretically confirmed that, when retrieval is successful, convergence is attained within several tens of iterations.

Figure 2 illustrates the overlap after 50, 100, and 200 iterations by color while the red solid lines represent the boundary of the basin of attraction assessed by an approximate dynamics discussed below. If the dynamics had fully converged, the region where the overlap remains finite would represent the basin of attraction, suggesting $\alpha'_{c,3} \simeq 0.33$. However, Fig. 1 shows a gradual decay of the overlap with increasing $t$, indicating that $\alpha'_{c,3}$ is at most about 0.3. Indeed, while the computer simulation results for $N = 512$ exhibit almost no dependence on $T$ due to finite-size effects (left panels of Fig. 2), the DMFT results shown in the right panels of Fig. 2 indicate a gradual shrinkage of the region with large $m$ as $T$ increases. Consistently, static analyses based on the replica method (Mézard et al., 1987) give smaller values $\alpha'_{c,3} \simeq 0.252$ under the replica symmetric ansatz, which is consistent with the approximate dynamics shown below, and $\alpha'_{c,3} \simeq 0.266$ under the 1-step replica symmetry breaking ansatz as shown in Appendix D. In similar systems, slow dynamics are known to occur near the phase boundary between the crystal and glassy phases (Krzakala & Zdeborová, 2011), which corresponds here to the retrieval success/failure transition. Therefore, the true basin of attraction is narrower than what is shown in Fig. 2, but we conjecture that accurately determining it is challenging due to the presence of slow dynamics. A similar situation arises for $n \geq 4$ as well.

## 5.2 CONNECTION TO RELATED ANALYSES

From the exact solution obtained via the generating functional analysis, we obtain the following approximated result when self-coupling is neglected.

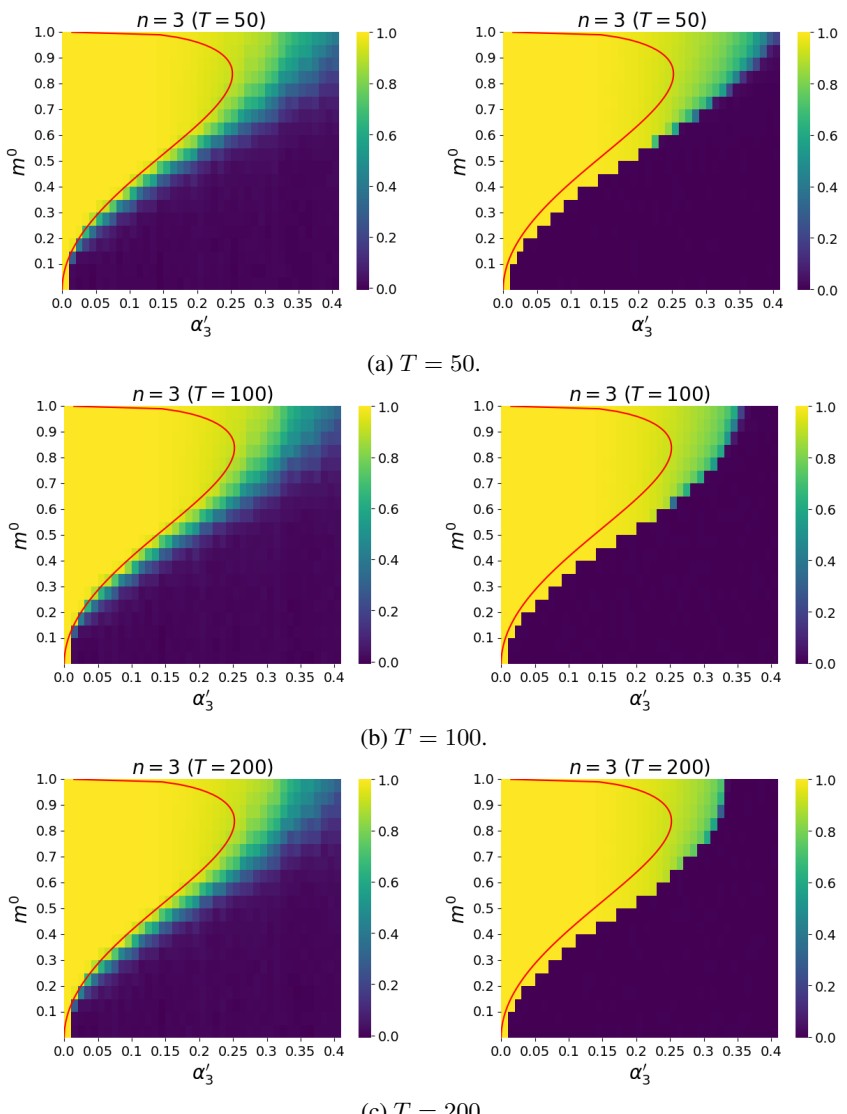

(a) $T = 50$.

(b) $T = 100$.

(c) $T = 200$.

Figure 2: Overlap with the retrieved pattern $m = \boldsymbol{\xi}^1 \cdot \boldsymbol{h}/N$ after $T = 50, 100, 200$ iterations for Krotov's dense associative memory with $F(x) = x^n$ and $n = 3$. Left: computer simulations, 100 trials, $N = 512$. Right: theory. Red lines: attraction basin obtained by the approximate dynamics discussed in Sec. 5.2.

**Corollary 1.** *Neglecting the retarded self-interaction term as an approximation, i.e., setting $\Gamma = O$, we obtain*

$$m^{(t+1)} = \mathrm{erf}\left( \frac{(m^{(t)})^{n-1}}{\sqrt{(2n-3)!!\, 2\alpha_n}} \right), \tag{25}$$

*where* $\mathrm{erf}(x) := \frac{2}{\sqrt{\pi}} \int_0^x e^{-z^2}\, dz$ *denotes the error function, and $m^{(0)}$ is the initial overlap.*

In this approximation, the equilibrium state of the dynamics can be simply obtained by setting $m^{(t)} = m$, and the resulting fixed-point equation corresponds to the equilibrium analysis by $n$-body Hopfield model. Although the coefficients differ, this is due to the fact that the energy function is not the same as that in Krotov's model. The storage capacity that obtained by the fixed-point equation of this approximated dynamics, i.e., $m = \mathrm{erf}(m^{n-1}/(\sqrt{(2n-3)!!\, 2\alpha_n}))$, gives that of the equilibrium analysis derived by the replica method.

We here consider the differences between Krotov's dense associative memory and the $n$-body Hopfield model independently proposed by Gardner and Abbott. The energy function of the $n$-body Hopfield model is defined by

$$H = -\frac{1}{\sqrt{2n!}N^{n-1}} \sum_{\mu=1}^{M} \sum_{j_1 \neq j_2 \neq \cdots \neq j_n}^{N} \xi_{j_1}^{\mu} \xi_{j_2}^{\mu} \cdots \xi_{j_n}^{\mu} \, h_{j_1} h_{j_2} \cdots h_{j_n}. \tag{26}$$

The values of the variables $j_1, \cdots, j_n$ are all distinct. This is the main difference from the Krotov's dense associative memory. Using the same way to Krotov's method, the corresponding update rule of the $n$-body Hopfield model is given as

$$h_i^{(t+1)} = \text{sgn} \left[ \sum_{\mu=1}^{M} \xi_i^{\mu} \frac{1}{N^{n-1}} \sum_{j_1 \neq \cdots \neq j_{n-1} \neq i}^{N} \xi_{j_1}^{\mu} \cdots \xi_{j_{n-1}}^{\mu} \, h_{j_1}^{(t)} \cdots h_{j_{n-1}}^{(t)} \right]. \tag{27}$$

We have the exact result in the same way to obtain Proposition 1. Let $M = \alpha_n N^{n-1}$ again. Neglecting the retarded self-interaction term as an approximation, i.e., setting $\Gamma = O$, we obtain

$$m^{(t)} = \text{erf} \left( \frac{(m^{(t-1)})^{n-1}}{\sqrt{(n-1)! \, 2\alpha_n}} \right). \tag{28}$$

The stationary equation, i.e., setting $m^{(t)} = m$, is equivalent to the result derived by Abbott (Abbott & Arian, 1987). The detail is available in Appendix C.

## 6 CONCLUSION

We performed an asymptotically exact analysis of the dynamical behaviour of dense associative memory using generating functional analysis (GFA) in the large-system limit. The analysis revealed the presence of a retarded self-coupling term, indicating that the next state of the system depends in a complex manner on all past states. We also confirmed that this property cannot be captured by a method based on the signal-to-noise analysis. In the traditional Hopfield model, i.e., $n = 2$, it was found that the system exhibits a noise variance that depends intricately on non-recalled patterns. In contrast, for $n \geq 3$, the covariance matrix due to non-recalled patterns has no explicit dependence on the overlap or the response function; it may depend on them only implicitly through the off-diagonal elements of the correlation function, which is determined self-consistently with the overlap and the response function. As a result, the recall process becomes simpler than the traditional Hopfield model.

Assuming the existence of a stationary state, we can also consider a macroscopic fixed-point equation from the GFA equations. Due to the presence of the self-coupling term, this result must differ from that of existing equilibrium analysis. This difference comes from the fact that, for models with $n \geq 3$, the system does not satisfy the detailed balance condition.

In this work, we provided an exact dynamical analysis of dense associative memory using the generating functional analysis, and verified the theoretical predictions with numerical experiments. Our results clarify how higher-order interactions, namely, $n \geq 3$, suppress the increasing of crosstalk noise due to the recalling pattern itself, thereby stabilizing recall dynamics and enhancing memory capacity. This contrasts with the classical Hopfield model, where self-retrieval inevitably introduces additional noise. These findings offer a quantitative framework to evaluate the stability and storage capacity of associative memory models, which is useful for guiding model design. While our experiments were limited to relatively small system sizes and specific interaction orders, the analytical methodology is general and can be applied to a broader class of energy-based models. This approach can be extended to modern Hopfield networks, memory-augmented architectures, and other energy-based formulations will provide further insights into the design of robust and scalable memory systems. In this context, the simplicial Hopfield networks can also be analyzed within the same framework. We are currently working on analyzing cases where the function $F$ introduced into the energy is exponential, as well as the case where memory patterns are biased (Bielmeier & Friedland, 2025).

ACKNOWLEDGMENTS

Useful discussions with Hajime Yoshino are gratefully acknowledged. This work was partially supported by JSPS KAKENHI Grant Nos. 23H05492, 25K24611 (KM, JT), 23K03841 (KM), 26K02981 (KM, YK) and MEXT/JSPS KAKENHI Grant No. 22H05117 (YK).

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

APPENDICES

## A    PROOF OF LEMMA 1

We first calculate the expectation value of the noise term of all non-recalled patterns, which is the last part in (14). Using the Taylor expansion, we obtain

$$
\mathbb{E}_{\boldsymbol{\xi}^2, \cdots, \boldsymbol{\xi}^M} \exp\left[-i \sum_{t=0}^{T-1} \sum_{i=1}^{N} \sum_{\mu=2}^{M} \hat{u}_i^{(t)} \xi_i^{\mu} \, n\left(\frac{1}{N} \sum_{j \neq i}^{N} \xi_j^{\mu} h_j^{(t)}\right)^{n-1}\right] \tag{29}
$$

$$
= \prod_{\mu=2}^{M} \mathbb{E}_{\boldsymbol{\xi}^{\mu}} \left\{ 1 + \frac{1}{2}\left(-i \sum_{t=0}^{T-1} \sum_{i=1}^{N} \hat{u}_i^{(t)} \xi_i^{\mu} \, n\left(\frac{1}{N} \sum_{j \neq i}^{N} \xi_j^{\mu} h_j^{(t)}\right)^{n-1}\right)^2 + O\left(\frac{n^3}{N^{3(n-1)}}\right)\right\} \tag{30}
$$

$$
= \exp\left[-\frac{n^2(M-1)}{2N^{2(n-1)}}\left(\sum_{t=0}^{T-1}\sum_{t'=0}^{T-1}\sum_{i=1}^{N} \hat{u}_i^{(t)}\hat{u}_i^{(t')}\mathcal{N}_1 + \sum_{t=0}^{T-1}\sum_{t'=0}^{T-1}\sum_{i=1}^{N}\sum_{i'\neq i}^{N}\hat{u}_i^{(t)}\hat{u}_{i'}^{(t')}\mathcal{N}_2\right) + O\left(\frac{n^3 M}{N^{3(n-1)}}\right)\right], \tag{31}
$$

where

$$
\mathcal{N}_1 = \mathbb{E}_{\boldsymbol{\xi}}\left[\sum_{j_1 \neq i}^{N} \cdots \sum_{j_{n-1} \neq i}^{N} \sum_{j'_1 \neq i'}^{N} \cdots \sum_{j'_{n-1} \neq i'}^{N} \xi_{j_1} \cdots \xi_{j_{n-1}} \xi_{j'_1} \cdots \xi_{j'_{n-1}} h_{j_1}^{(t)} \cdots h_{j_{n-1}}^{(t)} h_{j'_1}^{(t')} \cdots h_{j'_{n-1}}^{(t')}\right],
$$

$$
\mathcal{N}_2 = \mathbb{E}_{\boldsymbol{\xi}}\left[\xi_i \xi_{i'} \sum_{j_1 \neq i}^{N} \cdots \sum_{j_{n-1} \neq i}^{N} \sum_{j'_1 \neq i'}^{N} \cdots \sum_{j'_{n-1} \neq i'}^{N} \xi_{j_1} \cdots \xi_{j_{n-1}} \xi_{j'_1} \cdots \xi_{j'_{n-1}} h_{j_1}^{(t)} \cdots h_{j_{n-1}}^{(t)} h_{j'_1}^{(t')} \cdots h_{j'_{n-1}}^{(t')}\right].
$$

Since $\boldsymbol{\xi}^2, \cdots, \boldsymbol{\xi}^M$ are independent, we can drop the index $\mu$. It should be noted that any term in $\mathcal{N}_1$ and $\mathcal{N}_2$ that contains an odd number of identical index from the same pattern has zero expectation, because all $\xi_1, \cdots, \xi_N$ are independent and have zero mean.

We calculate $\mathcal{N}_1$. The leading term in $\mathcal{N}_1$ can be obtained by calculating the summations in the case where the $2(n-1)$ variables are grouped into pairs, each pair taking the same value. We have to do this for all possible partitions. We must distinguish three types of pairings: (i) between two primed variables, (ii) between two unprimed variables, and (iii) between a primed and an unprimed variable. Note that depending on the type of pair, the time parameter differs. Therefore, the leading term can be obtained by counting the number of ways to partition the $2(n-1)$ indices, i.e., $j_1, \cdots, j_{n-1}, j'_1, \cdots, j'_{n-1}$, into $n-1$ pairs in which indices take the same value while each different pairs takes different values.

We here consider two sets of indices: the set of unprimed indices $\mathcal{J} = \{j_1, \cdots, j_\ell\}$ and the set of primed indices $\mathcal{J}' = \{j'_1, \cdots, j'_\ell\}$. First, we consider the number of ways to divide $2\ell$ indices, including $\ell$ unprimed indices and $\ell$ primed indices, into $\ell$ pairs. Let $A(\ell, k)$ be the number of ways to have exactly $k$ unprimed-primed pairs in $\ell$ total pairs, which is given by

$$
A(\ell, k) = \binom{\ell}{k}^2 k! \, B(\ell - k)^2, \tag{32}
$$

where $B(m)$ is the number of ways where $m/2$ unprimed-unprimed pairs and $m/2$ primed-primed pairs are made using $2m$ indices, consisting of $m$ unprimed indices and $m$ primed indices:

$$
B(m) = \mathbf{1}_{m:\text{even}} \frac{1}{(m/2)!} \binom{m}{2}\binom{m-2}{2}\cdots\binom{2}{2} = \mathbf{1}_{m:\text{even}} \, (m-1)!!. \tag{33}
$$

Note that $\sum_{k=0}^{\ell} A(\ell, k) = B(2\ell)$ holds.

Using the quantity $A(\ell, k)$ and the identity $(h_i^{(t)})^2 = 1$, we obtain

$$\mathcal{N}_1 = \mathbb{E}_{\boldsymbol{\xi}} \left[ \sum_{j_1, \cdots, j_{n-1} \neq i} \sum_{j_1', \cdots, j_{n-1}' \neq i} \xi_{j_1} \cdots \xi_{j_{n-1}} \xi_{j_1'} \cdots \xi_{j_{n-1}'} h_{j_1}^{(t)} \cdots h_{j_{n-1}}^{(t)} h_{j_1'}^{(t')} \cdots h_{j_{n-1}'}^{(t')} \right] \tag{34}$$

$$= N^{n-1} \underbrace{\sum_{k=0}^{n-1} A(n-1, k) \left( \frac{1}{N} \sum_{j=1}^{N} h_j^{(t)} h_j^{(t')} \right)^k}_{=O(N^0)} + O(N^{n-3}). \tag{35}$$

Next, we calculate $\mathcal{N}_2$. For notational simplicity, let $( \cdot |_{j_1 = i'})(\cdots)$ be an operator to substitute $j_1 = i'$ into $(\cdots)$, and let $(\sum_{j_1 \neq i, \neq i'} \cdot )(\cdots)$ be an operator for summing $(\cdots)$ over $j_1 \neq i, \neq i'$. It should be noted that each of $j_1, \cdots, j_{n-1}$ can take the value $i'$, and conversely, each of $j_1', \cdots, j_{n-1}'$ can take the value $i$. For all $i \in \{1, \cdots, N\}$ and $i' \in \{1, \cdots, N\} \backslash \{i\}$, we have

$$\mathcal{N}_2 = \mathbb{E}_{\boldsymbol{\xi}} \Bigg[ \xi_i \xi_{i'} \left( \cdot \Big|_{j_1 = i'} + \sum_{j_1 \neq i, \neq i'} \cdot \right) \cdots \left( \cdot \Big|_{j_{n-1} = i'} + \sum_{j_{n-1} \neq i, \neq i'} \cdot \right)$$

$$\left( \cdot \Big|_{j_1' = i} + \sum_{j_1' \neq i', \neq i} \cdot \right) \cdots \left( \cdot \Big|_{j_{n-1}' = i} + \sum_{j_{n-1}' \neq i', \neq i} \cdot \right)$$

$$\xi_{j_1} \cdots \xi_{j_{n-1}} \xi_{j_1'} \cdots \xi_{j_{n-1}'} h_{j_1}^{(t)} \cdots h_{j_{n-1}}^{(t)} h_{j_1'}^{(t')} \cdots h_{j_{n-1}'}^{(t')} \Bigg] \tag{36}$$

$$= \binom{n-1}{1} \binom{n-1}{1} \mathbb{E}_{\boldsymbol{\xi}} \Bigg[ \xi_i \xi_{i'}$$

$$\left( \sum_{j_1 \neq i, \neq i'} \cdot \right) \cdots \left( \sum_{j_{n-2} \neq i, \neq i'} \cdot \right) \left( \cdot \Big|_{j_{n-1} = i'} \right)$$

$$\left( \sum_{j_1' \neq i', \neq i} \cdot \right) \cdots \left( \sum_{j_{n-2}' \neq i', \neq i} \cdot \right) \left( \cdot \Big|_{j_{n-1}' = i} \right)$$

$$\xi_{j_1} \cdots \xi_{j_{n-1}} \xi_{j_1'} \cdots \xi_{j_{n-1}'} h_{j_1}^{(t)} \cdots h_{j_{n-1}}^{(t)} h_{j_1'}^{(t')} \cdots h_{j_{n-1}'}^{(t')} \Bigg] + O(N^{n-4}) \tag{37}$$

$$= (n-1)^2 \ h_{i'}^{(t)} h_i^{(t')} \ \mathbb{E}_{\boldsymbol{\xi}} \Bigg[ \sum_{j_1, \cdots, j_{n-2} \neq i, \neq i'} \sum_{j_1', \cdots, j_{n-2}' \neq i', \neq i}$$

$$\xi_{j_1} \cdots \xi_{j_{n-2}} \xi_{j_1'} \cdots \xi_{j_{n-2}'} h_{j_1}^{(t)} \cdots h_{j_{n-2}}^{(t)} h_{j_1'}^{(t')} \cdots h_{j_{n-2}'}^{(t')} \Bigg] + O(N^{n-4}) \tag{38}$$

$$= (n-1)^2 \ h_{i'}^{(t)} h_i^{(t')} \ N^{n-2} \underbrace{\sum_{k=0}^{n-2} A(n-2, k) \left( \frac{1}{N} \sum_{j=1}^{N} h_j^{(t)} h_j^{(t')} \right)^k}_{=O(N^0)} + O(N^{n-4}). \tag{39}$$

Substituting (35) and (39) into (31), we obtain the expectation value of the noise term of all non-recalled patterns as follows:

$$\mathbb{E}_{\boldsymbol{\xi}^2, \cdots, \boldsymbol{\xi}^M} \exp\left[ -i \sum_{t=0}^{T-1} \sum_{i=1}^{N} \sum_{\mu=2}^{M} \hat{u}_i^{(t)} \xi_i^\mu \ n \left( \frac{1}{N} \sum_{j \neq i} \xi_j^\mu h_j^{(t)} \right)^{n-1} \right] \tag{40}$$

$$= \exp\left[ -\frac{1}{2} \cdot \frac{n^2 M}{N^{n-2}} \sum_{t=0}^{T-1} \sum_{t'=0}^{T-1} \Bigg\{ \right.$$

$$(n-1)^2 \left( \frac{1}{N} \sum_{i=1}^{N} h_i^{(t')} \hat{u}_i^{(t)} \right) \left( \frac{1}{N} \sum_{i'=1}^{N} h_{i'}^{(t)} \hat{u}_{i'}^{(t')} \right) \sum_{k=0}^{n-2} A(n-2, k) \left( \frac{1}{N} \sum_{j=1}^{N} h_j^{(t)} h_j^{(t')} \right)^k$$

$$+ \left( \frac{1}{N} \sum_{i=1}^{N} \hat{u}_i^{(t)} \hat{u}_i^{(t')} \right) \sum_{k=0}^{n-1} A(n-1,k) \left( \frac{1}{N} \sum_{j=1}^{N} h_j^{(t)} h_j^{(t')} \right)^k + O(N^{-1}) \right\} \right] \tag{41}$$

$$= \left[ \prod_{t=0}^{T} \prod_{t'=0}^{T} \int_{-\infty}^{\infty} dq^{(t,t')} \delta \left( N q^{(t,t')} - \sum_{i=1}^{N} h_i^{(t)} h_i^{(t')} \right) \right]$$

$$\times \left[ \prod_{t=0}^{T} \prod_{t'=0}^{T} \int_{-\infty}^{\infty} dQ^{(t,t')} \delta \left( N Q^{(t,t')} - \sum_{i=1}^{N} h_i^{(t)} \hat{u}_i^{(t')} \right) \right]$$

$$\times \left[ \prod_{t=0}^{T} \prod_{t'=0}^{T} \int_{-\infty}^{\infty} dK^{(t,t')} \delta \left( N K^{(t,t')} - \sum_{i=1}^{N} h_i^{(t)} \hat{u}_i^{(t')} \right) \right]$$

$$\times \exp \left[ -\frac{1}{2} \cdot \frac{n^2 M}{N^{n-2}} \sum_{t=0}^{T-1} \sum_{t'=0}^{T-1} \left\{ (n-1)^2 K^{(t,t')} K^{(t',t)} \sum_{k=0}^{n-2} A(n-2,k) \left( q^{(t,t')} \right)^k \right. \right.$$

$$\left. \left. + Q^{(t,t')} \sum_{k=0}^{n-1} A(n-1,k) \left( q^{(t,t')} \right)^k + O(N^{-1}) \right\} \right] \tag{42}$$

$$= \left[ \prod_{t=0}^{T} \prod_{t'=0}^{T} \int_{-\infty}^{\infty} dq^{(t,t')} \int_{-\infty}^{\infty} \frac{d\hat{q}^{(t,t')}}{2\pi} \exp \left\{ i\hat{q}^{(t,t')} \left( N q^{(t,t')} - \sum_{i=1}^{N} h_i^{(t)} h_i^{(t')} \right) \right\} \right]$$

$$\times \left[ \prod_{t=0}^{T} \prod_{t'=0}^{T} \int_{-\infty}^{\infty} dQ^{(t,t')} \int_{-\infty}^{\infty} \frac{d\hat{Q}^{(t,t')}}{2\pi} \exp \left\{ i\hat{q}^{(t,t')} \left( N Q^{(t,t')} - \sum_{i=1}^{N} h_i^{(t)} \hat{u}_i^{(t')} \right) \right\} \right]$$

$$\times \left[ \prod_{t=0}^{T} \prod_{t'=0}^{T} \int_{-\infty}^{\infty} dK^{(t,t')} \int_{-\infty}^{\infty} \frac{d\hat{K}^{(t,t')}}{2\pi} \exp \left\{ i\hat{q}^{(t,t')} \left( N K^{(t,t')} - \sum_{i=1}^{N} h_i^{(t)} \hat{u}_i^{(t')} \right) \right\} \right]$$

$$\times \exp \left[ -\frac{1}{2} \cdot \frac{n^2 M}{N^{n-2}} \sum_{t=0}^{T-1} \sum_{t'=0}^{T-1} \left\{ (n-1)^2 K^{(t,t')} K^{(t',t)} \sum_{k=0}^{n-2} A(n-2,k) \left( q^{(t,t')} \right)^k \right. \right.$$

$$\left. \left. + Q^{(t,t')} \sum_{k=0}^{n-1} A(n-1,k) \left( q^{(t,t')} \right)^k + O(N^{-1}) \right\} \right], \tag{43}$$

by using (15).

The signal term that includes the recalling pattern $\boldsymbol{\xi}^1$ can be rearranged as

$$\mathbb{E}_{\boldsymbol{\xi}^1} \exp \left[ -i \sum_{t=0}^{T-1} \sum_{i=1}^{N} \hat{u}_i^{(t)} \xi_i^1 \, n \left( \frac{1}{N} \sum_{j \neq i}^{N} \xi_j^1 h_j^{(t)} \right)^{n-1} \right]$$

$$= \mathbb{E}_{\boldsymbol{\xi}^1} \exp \left[ -iN \sum_{t=0}^{T-1} \left( \frac{1}{N} \sum_{i=1}^{N} \hat{u}_i^{(t)} \xi_i^1 \right) n \left( \frac{1}{N} \sum_{j=1}^{N} \xi_j^1 h_j^{(t)} + O(N^{-1}) \right)^{n-1} \right]$$

$$= \left[ \prod_{t=0}^{T} \int_{-\infty}^{\infty} dm^{(t)} \delta \left( N m^{(t)} - \sum_{i=1}^{N} \xi_i^1 h_i^{(t)} \right) \right]$$

$$\times \left[ \prod_{t=0}^{T} \int_{-\infty}^{\infty} dk^{(t)} \delta \left( N k^{(t)} - \sum_{i=1}^{N} \xi_i^1 \hat{u}_i^{(t)} \right) \right]$$

$$\times \mathbb{E}_{\boldsymbol{\xi}^1} \exp \left[ -iN \sum_{t=0}^{T-1} k^{(t)} \, n \left( m^{(t)} + O(N^{-1}) \right)^{n-1} \right] \tag{44}$$

$$= \left[ \prod_{t=0}^{T} \int_{-\infty}^{\infty} dm^{(t)} \int_{-\infty}^{\infty} \frac{d\hat{m}^{(t)}}{2\pi} \exp \left\{ i\hat{m}^{(t)} \left( N m^{(t)} - \sum_{i=1}^{N} \xi_i^1 h_i^{(t)} \right) \right\} \right]$$

$$\times \left[ \prod_{t=0}^{T} \int_{-\infty}^{\infty} dk^{(t)} \int_{-\infty}^{\infty} \frac{d\hat{k}^{(t)}}{2\pi} \exp\left\{ i\hat{k}^{(t)} \left( Nk^{(t)} - \sum_{i=1}^{N} \xi_i^1 \hat{u}_i^{(t)} \right) \right\} \right]$$

$$\times \mathbb{E}_{\boldsymbol{\xi}^1} \exp\left[ -iN \sum_{t=0}^{T-1} k^{(t)} n \left( m^{(t)} + O(N^{-1}) \right)^{n-1} \right]. \tag{45}$$

We here introduced the parameters of (15) using the Dirac delta function and its Fourier integral form of the Dirac delta function, i.e., $\delta(x) = \frac{1}{2\pi} \int_{-\infty}^{\infty} d\hat{x} e^{i\hat{x}x} = \frac{1}{2\pi i} \int_{-i\infty}^{i\infty} d\hat{x} e^{\hat{x}x}$. We then arrive at the generating functional of (16). Since the expectation over the non-recalled patterns has already been taken, and only the recalling pattern remains in the expression. The signal term of (45) is $e^{O(N)}$. On the other hand, the noise term of (43) is $e^{O(M/N^{n-2})}$. For non-trivial analysis, the signal term and the noise term must be of the same order, namely, the number of the memory patterns $M$ must be $O(N^{n-1})$.

# B  PROOF SKETCH OF PROPOSITION 1

It should be noted that the normalization relation $Z[\mathbf{0}] = 1$ plays an important role in the elimination of spurious solutions to the saddle-point equations. The terms in the averaged generating functional can be split into three related parts. The first one is a signal part. The second one is a static noise part due to the random variables within the model. The last one is retarded self-interaction due to the influence of the state at the previous stage, which may be able to affect the present state. The GFA allows us to treat the last part. After the analysis, it turns out that the system can be described in terms of the following three quantities:

$$m^{(t)} = \mathbb{E}_{\boldsymbol{\xi}^1, \cdots, \boldsymbol{\xi}^M} \left[ \left\langle \frac{1}{N} \sum_{i=1}^{N} \xi_i^1 h_i^{(t)} \right\rangle \right], \tag{46}$$

$$C^{(t,t')} = \mathbb{E}_{\boldsymbol{\xi}^1, \cdots, \boldsymbol{\xi}^M} \left[ \left\langle \frac{1}{N} \sum_{i=1}^{N} h_i^{(t)} h_i^{(t')} \right\rangle \right], \tag{47}$$

$$G^{(t,t')} = \mathbb{E}_{\boldsymbol{\xi}^1, \cdots, \boldsymbol{\xi}^M} \left[ \left\langle \frac{1}{N} \sum_{i=1}^{N} \frac{\partial h_i^{(t)}}{\partial \theta_i^{(t')}} \right\rangle \right], \tag{48}$$

where these are referred to as the *overlap*, the *correlation function*, and the *response function*, respectively. One can deduce the meaning of macroscopic parameters by differentiating the averaged generating functional with respect to the external field $\theta_i^{(t)}$ and generating functions $\psi_i^{(t)}$. The averaged generating functional $\bar{Z}[\boldsymbol{\psi}]$ is dominated by a saddle-point for $N \to \infty$. Using the normalization identity $\bar{Z}[\mathbf{0}] = \mathbb{E}_{\boldsymbol{\xi}^1, \cdots, \boldsymbol{\xi}^M} \langle 1 \rangle = 1$, one can have derivatives of the averaged generating functional:

$$\lim_{\boldsymbol{\psi} \to \mathbf{0}} \frac{\partial \bar{Z}[\boldsymbol{\psi}]}{\partial \psi_i^{(t)}} = -i \langle h^{(t)} \rangle_i,$$

$$\lim_{\boldsymbol{\psi} \to \mathbf{0}} \frac{\partial^2 \bar{Z}[\boldsymbol{\psi}]}{\partial \psi_i^{(t)} \partial \psi_{i'}^{(t')}} = -\delta_{i,i'} \langle h^{(t)} h^{(t')} \rangle_i - (1 - \delta_{i,i'}) \langle h^{(t)} \rangle_i \langle h^{(t')} \rangle_{i'},$$

$$\lim_{\boldsymbol{\psi} \to \mathbf{0}} \frac{\partial^2 \bar{Z}[\boldsymbol{\psi}]}{\partial \psi_i^{(t)} \partial \theta_{i'}^{(t')}} = -\delta_{i,i'} \langle h^{(t)} \hat{u}^{(t')} \rangle_i - (1 - \delta_{i,i'}) \langle h^{(t)} \rangle_i \langle \hat{u}^{(s')} \rangle_{i'} \tag{49}$$

$$\lim_{\boldsymbol{\psi} \to \mathbf{0}} \frac{\partial \bar{Z}[\boldsymbol{\psi}]}{\partial \theta_i^{(t)}} = -i \langle \hat{u}^{(t)} \rangle_i = 0$$

$$\lim_{\boldsymbol{\psi} \to \mathbf{0}} \frac{\partial^2 \bar{Z}[\boldsymbol{\psi}]}{\partial \theta_i^{(t)} \partial \theta_{i'}^{(t')}} = -\delta_{i,i'} \langle \hat{u}^{(t)} \hat{u}^{(t')} \rangle_i = 0,$$

where $\langle \; \rangle_i$ denotes the average that is defined by

$$\langle f(\boldsymbol{h}, \boldsymbol{u}, \hat{\boldsymbol{u}})\rangle_i := \frac{\displaystyle\sum_{\boldsymbol{h}} \int d\boldsymbol{u} d\hat{\boldsymbol{u}} \; w_i(\boldsymbol{h}, \boldsymbol{u}, \hat{\boldsymbol{u}}) f(\boldsymbol{h}, \boldsymbol{u}, \hat{\boldsymbol{u}})}{\displaystyle\sum_{\boldsymbol{h}} \int d\boldsymbol{u} d\hat{\boldsymbol{u}} \; w_i(\boldsymbol{h}, \boldsymbol{u}, \hat{\boldsymbol{u}})} \tag{50}$$

with

$$
\begin{aligned}
w_i(\boldsymbol{h}, \boldsymbol{u}, \hat{\boldsymbol{u}}) =& p[h^{(0)}]\left(\prod_{t=0}^{T-1} \delta[h^{(t+1)}; \mathrm{sgn}\,(u^{(t)})]\right) \\
& \times \exp\Bigg[-i\sum_{t=0}^{T-1}\sum_{t'=0}^{T-1}\{\hat{q}^{(s,s')}h^{(s)}\tilde{b}^{(s')} + \hat{Q}^{(s,s')}\hat{u}^{(s)}\hat{u}^{(s')} + \hat{K}^{(s,s')}h^{(s)}\hat{u}^{(s')}\} \\
& + i\sum_{t=0}^{T-1}\hat{u}^{(t)}\{u^{(t)} - \hat{k}^{(t)}\xi_i - \theta_i^{(t)}\} - i\sum_{t=0}^{T-1}h^{(s)}\hat{m}^{(t)}\Bigg]\Bigg|_{\mathrm{saddle}}.
\end{aligned}
\tag{51}
$$

The average $\langle(\cdots)\rangle_i$ is referred to as a *single-unit measure*. Here, evaluation $f|_{\mathrm{saddle}}$ denotes an evaluation of function $f$ at the dominating saddle-point. Substituting (49) into (10) – (12), we then have

$$
\begin{aligned}
&\mathbb{E}_{\boldsymbol{\xi}^1,\cdots,\boldsymbol{\xi}^M}\langle h_i^{(t)}\rangle = \langle h^{(t)}\rangle_i, \\
&\mathbb{E}_{\boldsymbol{\xi}^1,\cdots,\boldsymbol{\xi}^M}\langle h_i^{(t)} h_{i'}^{(t')}\rangle = \delta_{i,i'}\langle h^{(t)} h^{(t')}\rangle_i + (1-\delta_{i,i'})\langle h^{(t)}\rangle_i\langle h^{(t')}\rangle_{i'}, \\
&\mathbb{E}_{\boldsymbol{\xi}^1,\cdots,\boldsymbol{\xi}^M}\langle \frac{\partial h_i^{(t)}}{\partial\theta_{i'}^{(t')}}\rangle = -i\delta_{i,i'}\langle h^{(t)}\hat{u}^{(t')}\rangle_i.
\end{aligned}
\tag{52}
$$

In the large-system limit, the averaged generating functional will be evaluated by the dominating saddle-points of the exponent $\Phi + \Psi + \Omega$. We can now derive the saddle-point equations by differentiation with respect to the integral variables $m^{(t)}$, $\hat{m}^{(t)}$, $k^{(t)}$, $\hat{k}^{(t)}$, $q^{(t,t')}$, $\hat{q}^{(t,t')}$, $Q^{(t,t')}$, $\hat{Q}^{(t,t')}$, $K^{(t,t')}$, and $\hat{K}^{(t,t')}$. The saddle-point equations will involve the overlap $m^{(t)}$, the correlation $C^{(t,t')}$ and the response function $G^{(s,s')}$. It should be noted that causality, i.e.,

$$\frac{\partial\langle h^{(t)}\rangle}{\partial\theta^{(t')}} = 0, \tag{53}$$

should hold for $t \leq t'$. Therefore $G^{(t,t')} = 0$ for $t \leq t'$. Using causality and the identities (49) and (52), the straightforward differentiation of $\Phi + \Psi + \Omega$ with respect to the integral variables leads us to the following saddle-point equations:

$$m^{(t)} = \frac{1}{N}\sum_{i=1}^{N}\xi_i^1\overline{\langle h_i^{(t)}\rangle} = \langle\!\langle \xi h^{(t)}\rangle\!\rangle, \quad \hat{m}^{(t)} = 0, \quad k^{(t)} = 0, \quad \hat{k}^{(t)} = n(m^{(t)})^{n-1}, \tag{54}$$

$$q^{(t,t')} = C^{(t,t')} = \frac{1}{N}\sum_{i=1}^{N}\overline{\langle h_i^{(t)} h_i^{(t')}\rangle} = \langle\!\langle h^{(t)} h^{(t')}\rangle\!\rangle, \quad \hat{q}^{(t,t')} = 0, \tag{55}$$

$$Q^{(t,t')} = 0, \quad \hat{Q}^{(t,t')} = -i\frac{1}{2}R^{(t,t')}, \tag{56}$$

$$K^{(t,t')} = iG^{(t,t')} = \mathbf{1}_{t>t'}\frac{\partial\langle\!\langle h^{(t)}\rangle\!\rangle}{\partial\theta^{(t)}}, \quad \hat{K}^{(t,t')} = D^{(t,t')}G^{(t',t)}, \tag{57}$$

where

$$R^{(t,t')} = n^2\alpha_n\sum_{k=0}^{n-1}A(n-1,k)(C^{(t,t')})^k, \tag{58}$$

$$D^{(t,t')} = n^2(n-1)^2\alpha_n \sum_{k=0}^{n-2} A(n-2,k)(C^{(t,t')})^k. \tag{59}$$

Substituting the solutions of the saddle-point equation into the single-unit measure, we obtain the effective path measure. We then arrive at Proposition 1.

## C  PROOF SKETCH OF GARDNER'S MODEL

Using a similar way to Lemma 1, the expectation value of the noise term of all non-recalled patterns for the $n$-body Hopfield model is given by

$$
\begin{aligned}
&\mathbb{E}_{\boldsymbol{\xi}^2,\cdots,\boldsymbol{\xi}^M} \exp\left[-i\sum_{t=0}^{T-1}\sum_{i=1}^{N}\sum_{\mu=2}^{M}\hat{u}_i^{(t)}\xi_i^{\mu}\frac{1}{N^{n-1}}\sum_{j_1\neq\cdots\neq j_{n-1}\neq i}^{N}\xi_{j_1}^{\mu}\cdots\xi_{j_{n-1}}^{\mu}\, h_{j_1}^{(t)}\cdots h_{j_{n-1}}^{(t)}\right]\\
&=\exp\left[-\frac{1}{2}\frac{n^2 M}{N^{n-2}}\sum_{t=0}^{T-1}\sum_{t'=0}^{T-1}\left\{(n-1)^2\left(\frac{1}{N}\sum_{i=1}^{N}h_i^{(t')}\hat{u}_i^{(t)}\right)\left(\frac{1}{N}\sum_{i'=1}^{N}h_{i'}^{(t)}\hat{u}_{i'}^{(t')}\right)A(n-2,n-2)\right.\right.\\
&\quad\times\left.\left(\frac{1}{N}\sum_{j=1}^{N}h_j^{(t)}h_j^{(t')}\right)^{n-2}+\left(\frac{1}{N}\sum_{i=1}^{N}\hat{u}_i^{(t)}\hat{u}_i^{(t')}\right)A(n-1,n-1)\left(\frac{1}{N}\sum_{j=1}^{N}h_j^{(t)}h_j^{(t')}\right)^{n-1}+O(N^{-1})\right\}\right],
\end{aligned}
\tag{60}
$$

where

$$A(\ell,\ell) = \binom{\ell}{\ell}^2 \ell!\, B(\ell-\ell)^2 = \ell!. \tag{61}$$

Applying the same calculation, we arrive at (28).

## D  SKETCH OF REPLICA COMPUTATION

In a general setting, suppose that the state variable $\boldsymbol{s} = (s_i)$ is governed by a Hamiltonian $H(\boldsymbol{s}\mid\boldsymbol{r})$ that depends on a predetermined random variable $\boldsymbol{r}$. In this case, the thermal average

$$\langle\boldsymbol{s}\rangle = \frac{\mathrm{tr}_{\boldsymbol{s}}\,\boldsymbol{s}\,e^{-\beta H(\boldsymbol{s}|\boldsymbol{r})}}{Z(\boldsymbol{r})}, \qquad Z(\boldsymbol{r}) = \mathrm{tr}_{\boldsymbol{s}}\,e^{-\beta H(\boldsymbol{s}|\boldsymbol{r})}, \tag{62}$$

becomes a random quantity because it varies with the realization of $\boldsymbol{r}$. Here, $\mathrm{tr}_X(\cdots)$ denotes summation or integration over all possible configurations of $\boldsymbol{s}$.

The replica method is a technique used to evaluate the moments of the thermal average $\mathbb{E}_{\boldsymbol{r}}[\langle s_i\rangle^k]$ for $k = 1, 2, \ldots$, by means of the replica trick

$$\mathbb{E}_{\boldsymbol{r}}[\langle s_i\rangle^k] = \lim_{\nu\to 0}\frac{\mathbb{E}_{\boldsymbol{r}}\left[Z^{\nu}(\boldsymbol{r})\,\langle s_i\rangle^k\right]}{\mathbb{E}_{\boldsymbol{r}}[Z^{\nu}(\boldsymbol{r})]}. \tag{63}$$

In practice, this reduces the problem to computing the $\nu$-th moment of the partition function $\mathbb{E}_{\boldsymbol{r}}[Z^{\nu}(\boldsymbol{r})]$ for integers $\nu = 1, 2, \ldots$ using the saddle point method, and then analytically continuing the resulting expression to real values $\nu \in \mathbb{R}$ under the assumption of a certain symmetry.

For the model defined by (1) and (2), we analyze its behavior using the replica method under the assumption that $M = \alpha_n N^{n-1}$, and that only the overlap with the first pattern, $m = \boldsymbol{\xi}^1\cdot\boldsymbol{h}/N$, is $O(1)$ while the other overlaps, $\boldsymbol{\xi}^{\mu}\cdot\boldsymbol{h}/N$ for $\mu = 2,\ldots,\alpha_n N^{n-1}$, remain typically $O(N^{-1/2})$. To this end, we introduce the rescaled variables $u_{\mu} = \boldsymbol{\xi}^{\mu}\cdot\boldsymbol{h}/N^{1/2}$ for $\mu = 2,\ldots,\alpha_n N^{n-1}$, and rewrite the Hamiltonian as

$$H = -\frac{N}{2}m^n - \frac{N^{1-n/2}}{2}\sum_{\mu=2}^{\alpha_n N^{n-1}} u_{\mu}^n. \tag{64}$$

The corresponding partition function is thus given by

$$Z = \sum_{\boldsymbol{h}} \exp\left( \frac{N\beta m^n}{2} + \frac{N^{1-n/2}\beta}{2} \sum_{\mu=2}^{\alpha_n N^{n-1}} u_\mu^n \right), \tag{65}$$

and its $\nu$-th moment reads

$$\mathbb{E}_{\boldsymbol{\xi}}[Z^\nu] = \sum_{\boldsymbol{h}^1,\ldots,\boldsymbol{h}^\nu} \underbrace{\mathbb{E}_{\boldsymbol{\xi}^1}\left[ \exp\left( \frac{N\beta}{2} \sum_{a=1}^\nu (m^a)^n \right) \right]}_{\mathcal{I}_1} \times \underbrace{\mathbb{E}_{\boldsymbol{\xi}^2,\ldots,\boldsymbol{\xi}^{\alpha_n N^{n-1}}}\left[ \exp\left( \frac{N^{1-n/2}\beta}{2} \sum_{\mu=2}^{\alpha_n N^{n-1}} \sum_{a=1}^\nu (u_\mu^a)^n \right) \right]}_{\mathcal{I}_2}, \tag{66}$$

for natural numbers $\nu = 1, 2, \ldots$.

**Evaluation of $\mathcal{I}_2$.** The quantity $\mathcal{I}_2$ is evaluated using the following facts:

- The patterns $\boldsymbol{\xi}^2, \ldots, \boldsymbol{\xi}^{\alpha_n N^{n-1}}$ are independently drawn from the uniform distribution over $\{+1, -1\}^N$. Thus, $\mathcal{I}_2$ is obtained by averaging $\exp\left( \frac{N^{1-n/2}\beta}{2} \sum_{a=1}^\nu (u^a)^n \right)$ with respect to a single pattern $\boldsymbol{\xi}$ (i.e. dropping the subscript $\mu$), and raising the result to the power $\alpha_n N^{n-1}$.

- For $\boldsymbol{\xi}$ uniformly distributed over $\{+1, -1\}^N$, the central limit theorem implies that $u^1, \ldots, u^\nu$ follow a zero-mean multivariate normal distribution with covariance $\mathbb{E}_{\boldsymbol{\xi}}[u^a u^b] = N^{-1} \boldsymbol{h}^a \cdot \boldsymbol{h}^b =: q_{ab}$.

- For $n \geq 3$, the factor $N^{1-n/2}$ vanishes as $N \to \infty$. We therefore apply the Taylor expansion

$$\exp\left( \frac{N^{1-n/2}\beta}{2} \sum_{a=1}^\nu (u^a)^n \right)$$
$$= 1 + \frac{N^{1-n/2}\beta}{2} \sum_{a=1}^\nu (u^a)^n + \frac{1}{2}\left( \frac{N^{1-n/2}\beta}{2} \sum_{a=1}^\nu (u^a)^n \right)^2 + O(N^{3-3n/2}) \tag{67}$$

to compute the Gaussian average.

Using these observations, we obtain

$$\mathcal{I}_2 = \left( 1 + \frac{1}{2}\mathbb{E}_{u^1,\ldots,u^\nu}\left[ \left( \frac{N^{1-n/2}\beta}{2} \sum_{a=1}^\nu (u^a)^n \right)^2 \right] + O(N^{3-3n/2}) \right)^{\alpha_n N^{n-1}}$$
$$= \exp\left( \frac{N\alpha_n \beta^2}{8} \mathbb{E}_{u^1,\ldots,u^\nu}\left[ \sum_{a,b} (u^a u^b)^{2n} \right] + O(N^{2-n/2}) \right)$$
$$\simeq \exp\left( \frac{N\alpha_n \beta^2}{8} \mathbb{E}_{u^1,\ldots,u^\nu}\left[ \sum_{a,b} (u^a u^b)^{2n} \right] \right), \tag{68}$$

which is valid for $n \geq 3$.

**Evaluation of $\mathcal{I}_1$ and the subshell volume.** The contribution $\mathcal{I}_1$ is handled together with the volume of the subshell of configurations $\boldsymbol{h}^1, \ldots, \boldsymbol{h}^\nu$ that satisfy fixed order parameters $m^a$ and $q_{ab}$ $(a, b = 1, \ldots, \nu)$. Specifically, we insert the identity

$$1 \propto \int \prod_{a=1}^\nu dm_a \int \prod_{a<b} dq_{ab} \prod_{a=1}^\nu \delta(\boldsymbol{\xi}^1 \cdot \boldsymbol{h}^a - Nm^a) \prod_{a<b} \delta(\boldsymbol{h}^a \cdot \boldsymbol{h}^b - Nq_{ab}) \tag{69}$$

into (66). This leads to

$$\mathbb{E}_{\boldsymbol{\xi}^1}\left[\exp\left(\frac{N\beta}{2}\sum_{a=1}^{\nu}(m^a)^n\right)\prod_{a=1}^{\nu}\delta(\boldsymbol{\xi}^1\cdot\boldsymbol{h}^a - Nm^a)\right]$$

$$= \exp\left(\frac{N\beta}{2}\sum_{a=1}^{\nu}(m^a)^n\right) \times \mathbb{E}_{\boldsymbol{\xi}^1}\left[\prod_{a=1}^{\nu}\delta(\boldsymbol{\xi}^1\cdot\boldsymbol{h}^a - Nm^a)\right]. \tag{70}$$

Next, the subshell volume

$$\sum_{\boldsymbol{h}^1,\dots,\boldsymbol{h}^\nu}\prod_{a=1}^{\nu}\delta(\boldsymbol{\xi}^1\cdot\boldsymbol{h}^a - Nm^a)\prod_{a<b}\delta(\boldsymbol{h}^a\cdot\boldsymbol{h}^b - Nq_{ab}) \tag{71}$$

is evaluated using the Fourier representations

$$\delta(\boldsymbol{\xi}^1\cdot\boldsymbol{h}^a - Nm^a) = \frac{1}{2\pi i}\int_{-i\infty}^{i\infty}d\hat{m}_a\exp\left[\hat{m}_a(\boldsymbol{\xi}^1\cdot\boldsymbol{h}^a - Nm^a)\right], \tag{72}$$

$$\delta(\boldsymbol{h}^a\cdot\boldsymbol{h}^b - Nq_{ab}) = \frac{1}{2\pi i}\int_{-i\infty}^{i\infty}d\hat{q}_{ab}\exp\left[\hat{q}_{ab}(\boldsymbol{h}^a\cdot\boldsymbol{h}^b - Nq_{ab})\right]. \tag{73}$$

Combining all contributions and applying the saddle-point method, we finally obtain

$$\frac{1}{N}\ln\mathbb{E}_{\boldsymbol{\xi}}[Z^\nu] \simeq \operatorname*{extr}_{\{m_a,q_{ab},\hat{m}_a,\hat{q}_{ab}\}}\left\{\frac{\beta}{2}\sum_{a=1}^{\nu}(m^a)^n + \frac{\alpha_n\beta^2}{8}\mathbb{E}_{u^1,\dots,u^\nu}\left[\sum_{a,b}(u^au^b)^{2n}\right]\right.$$

$$-\sum_{a<b}\hat{q}_{ab}q_{ab} - \sum_{a=1}^{\nu}\hat{m}_am^a$$

$$\left. +\ln\mathbb{E}_{\xi}\left[\sum_{h^1,\dots,h^\nu}\exp\left(\sum_{a<b}\hat{q}_{ab}h^ah^b + \sum_{a=1}^{\nu}\hat{m}_a\xi h^a\right)\right]\right\}, \tag{74}$$

where $\operatorname{extr}_X\{f(X)\}$ generally stands for extremizing $f(X)$ with respect to $X$.

To proceed toward the limit $\nu\to 0$, we next impose an appropriate replica-symmetric (or symmetry-broken) ansatz for the saddle-point parameters.

## D.1 REPLICA SYMMETRIC SOLUTION

The replica-symmetric (RS) solution is obtained by imposing $m^a = m$, $q_{ab} = q$, $\hat{m}_a = \hat{m}$, $\hat{q}_{ab} = \hat{q}$ in (74). Under this ansatz, the Gaussian average becomes

$$\mathbb{E}_{u^1,\dots,u^\nu}\left[\sum_{a,b}(u^au^b)^{2n}\right] = \nu\, M_n(1) + \nu(\nu - 1)\, M_n(q), \tag{75}$$

where

$$M_n(\rho) = \sum_{r=0}^{\lfloor n/2\rfloor}\binom{n}{2r}(n - 2r)!\left((2r - 1)!!\right)^2\rho^{n-2r}. \tag{76}$$

Furthermore, using a Gaussian integral identity, we obtain

$$\mathbb{E}_{\xi}\left[\sum_{h^1,\dots,h^\nu}\exp\left(\hat{q}\sum_{a<b}h^ah^b + \hat{m}\sum_{a=1}^{\nu}\xi h^a\right)\right]$$

$$= e^{-\nu\hat{q}/2} \int Dz\, \mathbb{E}_{\xi} \left[ \left( 2\cosh\left( \sqrt{\hat{q}}\, z + \hat{m}\xi \right) \right)^{\nu} \right]$$

$$= e^{-\nu\hat{q}/2} \int Dz\, \left( 2\cosh\left( \sqrt{\hat{q}}\, z + \hat{m}\xi \right) \right)^{\nu}, \tag{77}$$

where $Dz = dz\, e^{-z^2/2}/\sqrt{2\pi}$ denotes the standard Gaussian measure.

To characterize the retrieval state—that is, a (local) minimum of the Hamiltonian $H$—we consider the zero-temperature limit $\beta \to \infty$. In this limit, we introduce the rescaled parameters

$$F = \beta^{-2}\hat{q}, \qquad K = \beta^{-1}\hat{m}, \qquad \chi = \beta(1 - q). \tag{78}$$

For the case $n = 3$, substituting these scalings into (74) yields

$$\frac{1}{N}\mathbb{E}_{\xi}\left[\min_{h} H\right] = -\lim_{\beta\to\infty} \frac{1}{\beta N}\mathbb{E}_{\xi}[\ln Z] = -\lim_{\beta\to\infty} \frac{1}{\beta N}\lim_{\nu\to 0}\frac{\partial}{\partial\nu}\ln\mathbb{E}_{\xi}[Z^{\nu}]$$

$$= -\underset{\{m,\chi,F,K\}}{\mathrm{extr}} \left\{ \frac{m^3}{2} + \frac{27\alpha_3\chi}{8} - \frac{F\chi}{2} - Km + \int Dz\, \left| \sqrt{F}\, z + K \right| \right\}. \tag{79}$$

The value of $m$ determined by the extremization means the typical overlap with the retrieved pattern $\mathbb{E}_{\xi}[\xi^1 \cdot h]/N$. After performing the extremization, we obtain the fixed-point equation

$$m = \mathrm{erf}\left( \frac{m^2}{\sqrt{6\alpha_3}} \right) = \mathrm{erf}\left( \frac{m^2}{\sqrt{2\alpha_3'}} \right), \tag{80}$$

which coincides with the fixed-point condition of the approximate algorithm given in (25) for $n = 3$.

## D.2   1-STEP REPLICA SYMMETRY BREAKING SOLUTION

Under the one-step replica-symmetry-breaking (1RSB) ansatz, the replica indices $1,\ldots,\nu$ are divided into $\nu/x$ groups, each of size $x$. The order parameters in (74) are set as

$$q_{ab} = \begin{cases} q_1, & \text{if } a \text{ and } b \text{ belong to the same group,} \\ q_0, & \text{otherwise,} \end{cases} \tag{81}$$

and similarly for $\hat{q}_{ab}$. For $m^a$ and $\hat{m}_a$, we retain the RS conventions $m^a = m$ and $\hat{m}_a = \hat{m}$.

Under this ansatz, the Gaussian average becomes

$$\mathbb{E}_{u^1,\ldots,u^{\nu}} \left[ \sum_{a,b} (u^a u^b)^{2n} \right] = \nu\, M_n(1) + \frac{\nu}{x}\, x(x-1)\, M_n(q_1) + x^2 \frac{\nu}{x}\left( \frac{\nu}{x} - 1 \right) M_n(q_0), \tag{82}$$

where $M_n(\rho)$ is defined in (76).

We also obtain

$$\mathbb{E}_{\xi} \left[ \sum_{h^1,\ldots,h^{\nu}} \exp\left( \hat{q}_{ab} \sum_{a<b} h^a h^b + \hat{m} \sum_{a=1}^{\nu} \xi h^a \right) \right]$$

$$= e^{-\nu\hat{q}_1/2} \int Dz\, \left[ \int Dy\, \left( 2\cosh\left( \sqrt{\hat{q}_1 - \hat{q}_0}\, y + \sqrt{\hat{q}_0}\, z + \hat{m} \right) \right)^x \right]^{\nu/x}. \tag{83}$$

As before, to characterize the retrieval state—a local minimum of the Hamiltonian $H$—we consider the zero-temperature limit $\beta \to \infty$. In this limit, we introduce the rescaled variables

$$F_1 = \beta^{-2}\hat{q}_1, \qquad F_0 = \beta^{-2}\hat{q}_0, \qquad K = \beta^{-1}\hat{m}, \qquad \chi = \beta(1 - q_1), \qquad \mu = \beta x. \tag{84}$$

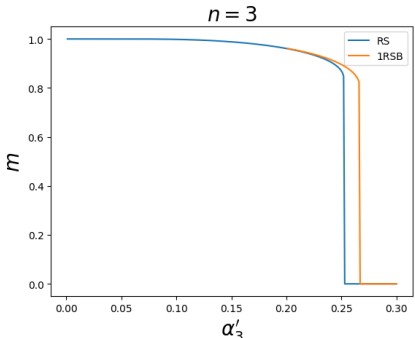

Figure 3: Typical values of overlap $m = \mathbb{E}_{\boldsymbol{\xi}}[\boldsymbol{\xi}^1 \cdot \boldsymbol{h}]/N$ evaluated under the RS and 1RSB ansatzes.

For the case $n = 3$, substituting these definitions into (74) yields

$$
\frac{1}{N}\mathbb{E}_{\boldsymbol{\xi}}\left[\min_{\boldsymbol{h}} H\right] = -\lim_{\beta \to \infty}\frac{1}{\beta N}\mathbb{E}_{\boldsymbol{\xi}}[\ln Z] = -\lim_{\beta \to \infty}\frac{1}{\beta N}\lim_{\nu \to 0}\frac{\partial}{\partial \nu}\ln \mathbb{E}_{\boldsymbol{\xi}}[Z^{\nu}]
$$

$$
= -\operatorname*{extr}_{\{m,\chi,q_0,F_1,F_0,K,\mu\}}\left\{\frac{m^3}{2} + \frac{\alpha_3}{8}\left[27\chi + \mu\left(9(1-q_0) + 6(1-q_0^3)\right)\right] - \frac{F_1\chi}{2} - \frac{\mu}{2}\left(F_1 - F_0 q_0\right) - Km\right.
$$

$$
\left. + \frac{1}{\mu}\int Dz \ln\left[\int Dy \exp\left(\mu \left|\sqrt{F_1 - F_0}\, y + \sqrt{F_0}\, z + K\right|\right)\right]\right\}. \quad (85)
$$

The RS solution corresponds to a special case of the 1RSB solution, characterized by the constraints

$$
q_1 = q_0 = q, \qquad \hat{q}_1 = \hat{q}_0 = \hat{q}.
$$

Hence, the local stability of the RS solution can be examined by linearizing the 1RSB extremum conditions with respect to the small perturbations

$$
\Delta q = q_1 - q_0, \qquad \Delta \hat{q} = \hat{q}_1 - \hat{q}_0,
$$

around the RS saddle point. This procedure yields the stability condition

$$
1 > 9\alpha_3\beta^2 q \int Dz \left[1 - \tanh^2\left(\sqrt{\hat{q}}\, z + \hat{m}\right)\right]^2
$$

$$
= 9\alpha_3 q \int Dz \left(\frac{\partial}{\partial K}\tanh\left(\beta\left(\sqrt{F}\, z + K\right)\right)\right)^2. \quad (86)
$$

However, this condition is never satisfied in the zero-temperature limit. Indeed, one finds

$$
\lim_{\beta \to \infty}\frac{\partial}{\partial K}\tanh\left(\beta\left(\sqrt{F}\, z + K\right)\right) = 2\,\delta\left(\sqrt{F}\, z + K\right),
$$

which causes the right-hand side of the stability condition (86) to diverge.

This demonstrates that the RS solution is always unstable at zero temperature, implying that replica-symmetry breaking must be taken into account in order to obtain a correct description of the model defined by (1) and (2).

### D.3 Solutions

We numerically performed the extremization of (79) and (85) for the case $n = 3$. The resulting values of $m$ are plotted versus $\alpha_3' = 3\alpha_3$ in Fig. 3. The figure shows that the storage capacity is estimated as

$$
\alpha_{c,3}' \simeq 0.252 \quad \text{under the RS ansatz}, \qquad \alpha_{c,3}' \simeq 0.266 \quad \text{under the 1RSB ansatz}.
$$

