# OpenReview forum: "Dynamical properties of dense associative memory"
_ICLR.cc/2026/Conference — ICLR 2026 Poster_

### Official Review · Reviewer_cF5v · 2025-10-26

**Soundness:** 3
**Presentation:** 2
**Contribution:** 3
**Rating:** 2
**Confidence:** 4

**Summary:**

The work provides an analysis of the dynamical properties of the Dense Associative Memory model (DAM), focusing on the high-order polynomial function ($n\geq 3$) from [Krotov and Hopfield (2016)](https://arxiv.org/abs/1606.01164) using generating functional analysis (GFA). The analysis of the work enables a quantitative evaluation of the convergence time and the storage capacity of DAM, shedding further light on why memory retrieval with DAM is more robust than the Classical Hopfield model.

**Strengths:**

1. An interesting, new work which studies the dynamical aspects of DAM in attempt to provide another perspective on the storage of capacity of these models.

2. The approximated or calculated dynamics are shown to be close to the simulated dynamics, illustrated in Fig. (2).

**Weaknesses:**

1. Limited experimentation. As far as I am aware, the experimentation is focused on the case of the polynomial order $n = 3$ while the mathematical derivations are focused on the case of $n \geq 3$.

2. The experiments are not conducted on natural datasets (e.g., CIFAR10, ImageNet, etc.) or a small subset of them. In such datasets, there exists a strong need to understand how the diversity of pattern-features affect the dynamical regimes of DAMs, which are governed by their memories, mathematically.

3. What about asynchronous update? This work focuses on parallel update, which technically had been shown by [Cheung et al. (1987)](https://opg.optica.org/ao/abstract.cfm?uri=ao-26-22-4808) that it does not guarantee convergence for $n = 2$, and it is not the preferred update in [Krotov and Hopfield (2016)].

**Questions:**

Overall, I think the work is interesting. But the presentation can be written much better, and it requires a lot more experimentation to help explain its mathematical contributions. See the weakness section above for my concerns/questions.

---

> ### Author Response · Authors · 2025-11-21
>
> We greatly appreciate the reviewers’ constructive review and insightful suggestions regarding our paper. Our responses are given in a point-by-point manner for each comment. We kindly ask you to also refer to the official comment on the overall manuscript, which appears at the top of this page.
>
>
> **W1.**
>
> The trends observed in the experiments and numerical analysis are similar also in the case of $n=4$. Since the number of storable patterns is $O(N^{,n-1})$ and therefore very large, computer simulations for sufficiently large $N$ become computationally demanding. We have added “A similar situation arises for $n \ge 4$ as well.” just before Section 5.2.
>
> **W2.**
>
> As you pointed out, extending the analysis to real datasets such as CIFAR-10 or ImageNet is an important future direction. In our current analysis, in order to handle a large number of stored patterns, we assume that each component of the stored patterns follows an independent Bernoulli distribution. Investigating probability distributions that more closely reflect real datasets, as well as developing a theoretical treatment for such cases, are important future challenges, and we would very much like to pursue these directions.
>
> **W3.**
>
> We sincerely appreciate this important comment. As you correctly noted, asynchronous updating is originally used in Krotov’s model, and handling asynchronous updates is theoretically important. In principle, the theoretical results for the asynchronous case can be derived; however, at present, numerical methods for evaluating these theoretical results have not yet been developed. For this reason, in this paper we restricted our analysis to the parallel update rule. Developing both the analysis and numerical evaluation methods for the asynchronous update case is highly important, and we are currently working on this problem.

---

> ### Comment · Reviewer_cF5v · 2025-11-25
>
> Thank you for your response. I am happy to increase my score from 2 -> 6. I do think the proofs are cleanly written and it is clear enough for me to read. However, I did mostly judge the work based on its experimentation rather than the proofs. Also, I do have a few suggestions:
>
> 1. For Fig. (1), please make your x and y labels larger alongside with your x and y tick values.
>
> 2. In Fig. (2), could you provide further ablations? What if you reduce T = 50, 100? Is it possible to see the same plot for $n = 4$ and $n = 5$?
>
> Also, another interesting work which you could cite is this [one](https://openreview.net/attachment?id=sbVZiZmfZ4&name=pdf).

---

> > ### Author Response · Authors · 2025-11-30
> >
> > We greatly appreciate your constructive and insightful comments. Our responses are given in a point-by-point manner for each comment.
> >
> > **C1.**
> >
> > As you pointed out, the numerical values in the figures were difficult to read, so we enlarged the labels and numbers in Fig. 1.
> >
> > **C2.**
> >
> > We sincerely appreciate this important comment. As you noted, it is important to examine the attraction basins at $T=50, 100$ in order to observe the slow dynamics. We have added the basins of attraction for $T=50, 100$ to Fig. 2. The numerical experiments were conducted with $N=512$ units. At this scale, it becomes difficult for the simulations to capture the very slow temporal changes. We believe that the fact that the attraction basin obtained from the numerical experiments change little even when $T$ varies is due to finite-size effects. In addition, we have added an explanation “Indeed, while the computer simulation results for $N=512$ exhibit almost no dependence on $T$ due to finite-size effects (left panels of Fig. 2), the DMFT results shown in the right panels of Fig. 2 indicate a gradual shrinkage of the region with large $m$ as $T$ increases.”  at the fifth line of the last paragraph in Section 5.1 (the 440th line). We are also considering producing similar graphs for $n=4, 5$. As $n$ increases, the memory capacity becomes larger, so we are carefully proceeding to prepare graphs. If we can complete the analysis in time for the manuscript revision deadline, we will update the manuscript again.
> >
> > **Comment under C2.**
> >
> > Thank you very much for your important comments. We also consider it essential to analyze cases in which memory patterns are correlated, and we are currently investigating situations where the patterns exhibit bias. We share the same motivation as the paper you pointed out, and we have revised the last sentence of the Conclusion: "We are currently working on analyzing cases where the function $F$ introduced into the energy is exponential, as well as the case where memory patterns are biased (Bielmeier & Friedland, 2025)."

---

### Official Review · Reviewer_V2XS · 2025-10-28

**Soundness:** 4
**Presentation:** 4
**Contribution:** 4
**Rating:** 10
**Confidence:** 4

**Summary:**

I feel like I'm slightly cheating here, since I'm somewhat plagiarizing the last paragraph of the paper. But that was an almost perfect summary, and it's more or less what I wanted to say anyway. At least I didn't use an LLM. ;)

The authors provide an analysis of the time evolution of statistical quantities (means, covariances, etc.) of the dense associative memory. For that they used the generating functional analysis, which is typically exact in the limit N --> infinity. (I'm guessing there are some cases where it breaks down, but I don't know what they are.) They veriﬁed the theoretical predictions with numerical experiments.

**Strengths:**

The calculation itself (which I only spot-checked) was highly nontrivial. While the technique itself is not knew, to my knowledge this is the first time it was applied to the dense associattive memory. I can also verify the statement by the author that their results cannot be captured by the standard signal-noise analysis (or at least not in my hands).

The results shed new light on the dynamics for cubic and higher nonlinearities and higher, and some interesting phenomena emerged -- such as suppression of crosstalk due to the recalled pattern, and a complete characterization of the basin of attraction.

The authors verified their theoretica predictions with numerical simulations, and agreement was good even though they used only 512 neurons. This is impressive, given that deviations usually scale as 1/sqrt(N), which in their case is 5%.

**Weaknesses:**

I didn't see any weaknesses. Unless you count a possible typo: it looks to me like theta_i is in a different place in Eq. 13 versus Eq. 6. But maybe that's on purpose?

**Questions:**

No questions.

---

> ### Author Response · Authors · 2025-11-21
>
> We greatly appreciate the reviewer's constructive comments and positive evaluation. We kindly ask you to also refer to the official comment on the overall manuscript, which appears at the top of this page.
>
> **W1.**
>
> We appreciate this important remark. This was a typo in Eq. (6), which has now been corrected.

---

### Official Review · Reviewer_oz6i · 2025-10-29

**Soundness:** 4
**Presentation:** 3
**Contribution:** 4
**Rating:** 8
**Confidence:** 4

**Summary:**

Using Generating Functional Analysis (GFA), the authors derived an asymptotically exact solution that quantitatively characterises convergence time and the size of attraction basins, aspects previously under-investigated. The analysis provides novel insight into the model's robustness, showing that unlike the traditional model, pattern retrieval does not introduce additional self-noise.

**Strengths:**

1. The work provides an asymptotically exact analysis of the dense associative memory's dynamics in the large-system limit using GFA.
2. The analysis yields explicit and quantitative results on convergence properties, such as the convergence time and the size of the attraction basins.
3. The modern/dense model is shown to be more robust than the traditional model because the noise variance does not depend on the overlap and does not increase as the overlap grows.

**Weaknesses:**

1. Rather than providing full proofs, proof sketches are provided. I can see and understand most of the results, however it would be ideal to have full, rigorous proofs provided.
2. There are some minor typos, e.g., "confirms" -> "confirmed" in L393. I suggest the authors proofread carefully.

**Questions:**

1. Could the authors' methods be simply extended to related formulations or models, e.g., Burns & Fukai "Simplicial Hopfield networks" ICLR 2023. (Probably it works similarly to the proof sketch of Gardner's model.)

---

> ### Author Response · Authors · 2025-11-21
>
> We greatly appreciate the reviewers’ constructive review and insightful suggestions regarding our paper. Our responses are given in a point-by-point manner for each comment. We kindly ask you to also refer to the official comment on the overall manuscript, which appears at the top of this page.
>
>
> **W1.**
>
> Thank you for your comment. As you pointed out, the text was indeed somewhat difficult to read.We have added further details to the derivation in Appendix A.
>
> **W2.**
>
> Thank you for pointing this out. We have checked and corrected the typos.
>
> **Q1.**
>
> Thank you very much for suggesting this highly interesting direction for extending the theory. Simplicial Hopfield networks correspond to the case where the function $F$ in the energy function (1) depends on the pattern index $\mu$. We believe that the same theoretical framework can be applied to analyze their dynamical properties. At the end of the conclusion section, we will add the sentence: “In this context, the simplicial Hopfield networks can also be analyzed within the same framework.”

---

> > ### Author Response · Authors · 2025-11-30
> >
> > **Supplement for our reply on W1.**
> >
> > We noticed that there were typos in the added text, so we have corrected it and added the detail in Appendix A.

---

### Official Review · Reviewer_f4Gk · 2025-10-31

**Soundness:** 3
**Presentation:** 3
**Contribution:** 3
**Rating:** 8
**Confidence:** 4

**Summary:**

The paper provides the first asymptotically exact dynamical analysis of dense associative memory (DAM) introduced by Krotov & Hopfield (2016), using the Generating Functional Analysis (GFA) framework.
While previous works mostly studied stationary-state storage capacity, this work studies on dynamical properties like recall convergence time, attraction basin region and self-noise behavior.
The main results is DAMs exhibit retarded self-interaction that stabilizes dynamics and prevents the recall process from generating additional self-noise. This makes memory retrieval more robust and expands the basin of attraction.

**Strengths:**

Overall, the paper is of high quality, I believe it provides a really good analysis on the whole recall process which previous studies did not fully explore.

1. The motivation is clear
2. The mathematical derivation looks correct to me
3. The results are novel to my best knowledge

**Weaknesses:**

1. The term "retarded self-interaction" lacks intuition to me, can the authors provide details on how do we interpret this term biologically?
2. It would be interesting to see whether the framework used in the paper can be easily expanded to different non-linear activation such as the exponential function.

**Questions:**

1. Will the argument $M=O(N^{n−1})$ become $M = O(e^N)$, if we switch the activation function to e^x?
2. Can the authors further discuss the limitations of this work, and potential future directions?
3. I suggest the authors to provide a table to compare existing results and their result.
4. Can the authors explain why the existing framework under control theory not work here? Based on my understanding, previous studies shows that the DAM system typically has exponential convergence rate as long as its stable. How does the result from this paper differ from this conclusion?

---

> ### Author Response · Authors · 2025-11-21
>
> We greatly appreciate the reviewers’ constructive review and insightful suggestions regarding our paper. Our responses are given in a point-by-point manner for each comment. We kindly ask you to also refer to the official comment on the overall manuscript, which appears at the top of this page.
>
>
> **W1.**
>
> In Krotov's model, as shown in Eq. (3) in the manuscript, the state $h_i^{(t)}$ of the $i$-th unit at time $t$ is not used to determine the state $h_i^{(t+1)}$ of the same unit at time $t+1$. Therefore, there is no self-coupling. However, the state $h_i^{(t)}$ is used to determine the states of all units other than the $i$-th one at time $t+1$. Consequently, the state $h_i^{(t)}$ influences its own future states $h_i^{(t+2)}, h_i^{(t+3)}, \cdots$ through all the other units after two steps and later. In the manuscript, we refer to this as "retarded self-interaction." To clarify this point, we added the following sentence three lines below Eq. (24): "The retarded self-interaction means the magnitude of the influence that returns to a unit itself after propagating through other units."
>
>
> **W2.**
>
> We conduct an asymptotic analysis, and therefore we can treat a wide variety of activation functions by focusing on the leading terms, as in Eq. (4). It should be noted that in the case of the exponential function, when the function $F$ in Eq. (4) is expanded into a power series, all terms have the same order with respect to the number of units $N$. Thus, an analysis that takes into account all the higher-order terms is required. We are currently working on this issue. Three lines below Eq. (4) in the manuscript, we have added the following explanation: “By focusing on the leading terms for a given function $F$, we can treat arbitrary activation functions. However, note that in the case of the exponential function, all terms in the power-series expansion have the same order.” We have also added the following sentence at the end of the conclusion section: “We are now working on the analysis of the case where the function $F$ introduced in the energy is exponential.“
>
> **Q1.**
>
> We are currently examining the case of $F(x) = e^{x}$ as one of our future works. The order of the number of stored patterns $M$ is determined by the order of the quantity in Eq. (29) in Appendix A, that is, the magnitude of the interference originating from patterns other than the recalling pattern. In the case of $F(x) = e^{x}$, when $F$ is expanded into a power series, all the terms have the same order, and therefore it is expected that Eq. (29) will include contributions not only from the $(n-1)$-th order term but from all orders. For this reason, we are now thinking that the number of storable patterns will be exponential.
>
> **Q2.**
>
> The analysis for the case $F(x) = e^{x}$, which you pointed out, is an important direction for future work. We also believe that the Hopfield layer can be analyzed using the same approach. Although it is empirically known that the Hopfield layer converges rapidly, the number of iterations required for convergence is expected to depend on the number of stored patterns. We are therefore considering investigating such properties. In addition, by examining the effects of sparsifying or discretizing the stored information, we believe that this line of research may also contribute to improving the efficiency of large language models.
>
> **Q3.**
>
> As you pointed out, there have been many important results on the dynamics of associative memory in previous research. We have summarized the representative analysis in Table 1.
>
> **Q4.**
>
> In the existing analyses, the retarded self-interaction cannot be treated, and therefore the resulting analysis is effectively equivalent to assuming that the current state is uncorrelated with the past states. In the analysis based on the path-integral method as well, if the retarded self-interaction is neglected -- that is, if the matrix $\Gamma$ appearing in Eq. (22) is set to the zero matrix $O$ -- then Eq. (25) is obtained. This expression shows exponential convergence.

---

### Author Response · Authors · 2025-11-21

In considering the possibility of adding further computational experiments, we noticed that, for $n \ge 3$ near the storage capacity, a "slow dynamics" phenomenon emerges—namely, when retrieval fails, the system requires a very long time to converge because the energy landscape becomes highly rugged. Because the dynamics take a very long time to settle, the basin of attraction appears larger than it actually is. When analyzing the equilibrium state using methods from equilibrium statistical mechanics, we found that the resulting storage capacity is smaller than the value estimated from the overlap at 200 iterations. To explain this point, we have added the following paragraph at the end of Section 5.1: “Figure 2 illustrates the overlap after 200 iterations by color while the red solid lines represent the boundary of the basin of attraction assessed by an approximate dynamics discussed below. If the dynamics had fully converged, the region where the overlap remains finite would represent the basin of attraction, suggesting $\alpha_{\mathrm{c}, 3}^\prime \simeq 0.33$. However, Fig. 2 shows a gradual decay of the overlap with increasing $t$, indicating that $\alpha_{\mathrm{c}, 3}^\prime$ is at most about $0.3$. On the other hand, static analyses based on the replica method (Mezard et al, 1987) give smaller values $\alpha_{\mathrm{c}, 3}^\prime \simeq 0.252$ under the replica symmetric ansatz, which is consistent with the approximate dynamics shown below, and $\alpha_{\mathrm{c}, 3}^\prime \simeq 0.266$ under the 1-step replica symmetry breaking ansatz as shown in Appendix D. In similar systems, slow dynamics are known to occur near the phase boundary between the crystal and glassy phases (Krzakala \& Zdeborova, 2011), which corresponds here to the retrieval success/failure transition. Therefore, the true basin of attraction is narrower than what is shown in Fig. 2, but we conjecture that accurately determining it is challenging due to the presence of slow dynamics. A similar situation arises for $n \ge 4$ as well.”

---

### Meta-Review · Area_Chair_kL2C · 2026-01-05

**Summary:**

This paper provides an exact dynamical analysis of dense associative memory, a modern Hopfield network variant, using generating functional methods. While prior work focused on stationary storage capacity, the authors characterize key dynamical properties of retrieval, including convergence time and attraction basin size, enabling a quantitative understanding of performance and guiding principled model design.

A main contribution is that, unlike classical Hopfield networks, self-retrieval does not induce additional noise, highlighting enhanced robustness arising from the modern network structure. Beyond dense associative memory, the proposed analytical framework applies broadly to energy-based models, making this work both technically strong and impactful for future architecture development.

**Reviewer Concerns:**

Nothing specific

**Reviewer Scores:**

can't predict

---

### Decision · Program_Chairs · 2026-01-26

Accept (Poster)